# Orbital-resolved observation of singlet fission

Alexander Neef[1✉], Samuel Beaulieu[1,2], Sebastian Hammer[3,7], Shuo Dong[1], Julian Maklar[1], Tommaso Pincelli[1,4], R. Patrick Xian[1,5], Martin Wolf[1], Laurenz Rettig[1], Jens Pflaum[3,6] & Ralph Ernstorfer[1,4✉]

Singlet fission[1–13] may boost photovoltaic efficiency[14–16] by transforming a singlet exciton into two triplet excitons and thereby doubling the number of excited charge carriers. The primary step of singlet fission is the ultrafast creation of the correlated triplet pair[17]. Whereas several mechanisms have been proposed to explain this step, none has emerged as a consensus. The challenge lies in tracking the transient excitonic states. Here we use time- and angle-resolved photoemission spectroscopy to observe the primary step of singlet fission in crystalline pentacene. Our results indicate a charge-transfer mediated mechanism with a hybridization of Frenkel and charge-transfer states in the lowest bright singlet exciton. We gained intimate knowledge about the localization and the orbital character of the exciton wave functions recorded in momentum maps. This allowed us to directly compare the localization of singlet and bitriplet excitons and decompose energetically overlapping states on the basis of their orbital character. Orbital- and localization-resolved many-body dynamics promise deep insights into the mechanics governing molecular systems[18–20] and topological materials[21–23].

The exciton doubling resulting from singlet fission (SF) could be harnessed for third-generation solar cells[14]. Einzinger et al. have shown that silicon solar cells can be sensitized with the SF material tetracene to harvest high-energy photons more efficiently[16]. It is generally accepted that SF proceeds in many steps, in which the primary step is the spin-allowed formation of the bitriplet exciton $^1$TT from the singlet exciton $S_1$ (Fig. 1a)[2]. The bitriplet can then spatially separate by triplet hopping and evolve into the spin-coherent but spatially separated bitriplet exciton $^1$T···T[24,25]. In the last step of SF, the spins of $^1$T···T eventually dephase and two independent triplets $T_1+T_1$ emerge. Intense efforts have been invested in explaining the primary step, the formation of $^1$TT. However, even in the most studied and highly efficient SF system, pentacene, ambiguity prevails over its mechanism.

Berkelbach et al. proposed a mechanism based on delocalized charge-transfer (CT) states[26], that is, excitons in which electron and hole are located on adjacent chromophores. They are contrasted with localized Frenkel states, in which electron and hole are confined to the same chromophore (Fig. 1a). In the CT-mediated mechanism, CT states are mixed into $S_1$ and accelerate SF by coupling strongly to both the bright Frenkel state and the dark bitriplet state. Owing to the energetic alignment of the CT states in pentacene, Berkelbach et al. conjectured that the bitriplet is populated by the decay of the singlet in a single step. In systems that show intramolecular SF, the CT-mediated mechanism has been substantially validated in solution[27–29].

In contrast to the CT-mediated mechanism, Chan et al. postulated a coherent mechanism[30]. They interpreted the instantaneous population of a low-energy signal in two-photon photoemission (PE) spectroscopy

as a signature of the bitriplet[3]. In the coherent mechanism, the bitriplet is formed quasi-instantaneously, which requires strong electronic coupling between diabatic Frenkel and bitriplet states[9,17].

Furthermore, an explanation based on a conical intersection mechanism has emerged, in which a vibrational mode shuttles the singlet exciton wave packet to a degeneracy of singlet and bitriplet manifolds, thereby driving SF[6,7,12]. In pentacene, an observed nuclear coherence has been interpreted as a signature of a conical intersection mechanism[7,12].

The ambiguity in matching these mechanisms to experimental observations stems from an assignment problem. In the methods used thus far, states are assigned by their energetic position. By going beyond spectral assignment and obtaining orbital- and localization-resolved information, we provide a more definite picture. We hence study SF in crystalline pentacene with time- and angle-resolved PE spectroscopy (trARPES). The recorded 4D PE intensity is proportional to the time-dependent state occupation $f(E, \Delta t)$, the dipole matrix element $M^{\mathbf{k}}_{f,i}$ and the wave function overlap between the impulsively ionized initial state ($i$) and the several possible cationic final states ($m$) of the $N-1$ electron system $\langle \Psi^{N-1}_m | \Psi^{N-1}_i \rangle$[31]:

$$I(\mathbf{k}, E, \Delta t) \propto f(E, \Delta t)|M^{\mathbf{k}}_{f,i}|^2 \sum_m |\langle \Psi^{N-1}_m | \Psi^{N-1}_i \rangle|^2 \delta(E + E^{N-1}_m - E^N_i - h\nu),$$

where the last term ensures energy conservation. The matrix element encodes information about the wave function of the initial state, which is imprinted in momentum maps, that is, constant energy cuts through the PE intensity. In the commonly used plane-wave approximation, the

[1]Department of Physical Chemistry, Fritz Haber Institute of the Max Planck Society, Berlin, Germany. [2]CELIA, University of Bordeaux-CNRS-CEA, Bordeaux, France. [3]Experimental Physics VI, Julius-Maximilian University Wuerzburg, Wuerzburg, Germany. [4]Institute for Optics and Atomic Physics, Technical University Berlin, Berlin, Germany. [5]Department of Statistical Sciences, University of Toronto, Toronto, Ontario, Canada. [6]Barvarian Centre for Applied Energy Research, Wuerzburg, Germany. [7]Present address: Center for the Physics of Materials, Departments of Physics and Chemistry, McGill University, Montréal, Quebec, Canada. ✉e-mail: neef@fhi.mpg.de; ernstorfer@tu-berlin.de

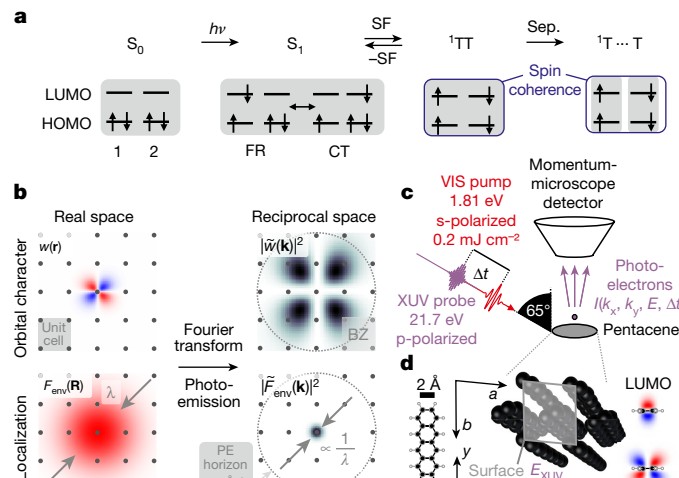

**Fig. 1 | Singlet fission, experimental principle and setup. a**, The SF reaction scheme. The electronic structures of the states in a HOMO–LUMO basis are indicated for a simplified two-chromophore (1, 2) SF system. **b**, An illustration of momentum maps of extended wave functions. PE essentially performs a Fourier transform on the wave function. Each factor to the wave function produces a distinct signature in the momentum maps recorded by trARPES. For clarity, the effect of the light polarization is neglected. **c**, Experimental setup. **d**, Molecular model, surface crystal structure and frontier orbitals of pentacene. $a = 6.26$ Å, $b = 7.79$ Å.

matrix element can be treated as a Fourier transform of the initial wave function[32–34]. The initial wave function in the solid state in turn can be cast as a sum of Wannier functions $w$ centred at the lattice sites $\mathbf{R}$ and multiplied by a phase factor:

$$\Psi_{\mathbf{k}}(\mathbf{r}) = \sum_{\mathbf{R}} w(\mathbf{r} - \mathbf{R}) F_{env}(\mathbf{R}) e^{i\mathbf{k}\cdot\mathbf{R}}.$$

Here we further introduced the phenomenological envelope function[35] $F_{env}(\mathbf{R})$, which is a distribution function centred at the origin. It modulates the magnitudes of Wannier functions on different unit cells and thereby their contribution to the total wave function.

We treat the PE process as a Fourier transform that converts the wave function in real space to a momentum map in reciprocal space (Fig. 1b). Each factor to the solid-state wave function distinctly visible in a momentum map shows a different piece of information. The Wannier function $w(\mathbf{r})$ encodes the orbital character and the envelope function $F_{env}(\mathbf{R})$ the localization of the wave function. In general, the electrons in molecular semiconductors are localized to several unit cells due to the weak intermolecular interaction and the presence of disorder[36,37]. As both the size of $F_{env}$ and $w$ are the same scale as the unit cell in these systems, both localization and orbital character should be observable in momentum maps. Here, we use these signatures of the wave function in momentum maps to distinguish the transient and energetically overlapping states in SF. We thereby obtain an orbital-resolved video of the SF process.

In the experiment, pentacene is excited with a 1.81 eV pump pulse polarized along the crystal $a$ axis, thus resonantly populating the lowest bright singlet exciton. The non-equilibrium state is then probed by a p-polarized ionizing extreme-ultraviolet (XUV) probe pulse (Fig. 1c,d)[38] with a system response function <50 fs. Figure 2a shows snapshots of the 3D PE intensity at three important time steps: before and directly after excitation, and after the primary step. In pentacene, each band features two branches due to the two inequivalent molecules in the unit cell. The dispersion of the well-separated electronic bands is apparent over the several Brillouin zones (BZ, $k_{BZ} \cong 0.9$ Å$^{-1}$, compare Extended Data Fig. 1a) within our momentum range ($k_{max} \approx 2.0$ Å$^{-1}$). The separation of the two valence band (VB) branches reaches its maximum $500 \pm 20$ meV at the M points (Extended Data Fig. 1b)[39]. Directly after excitation ($\Delta t = 0$ fs), two distinct excited-state signals simultaneously appear, the singlet exciton feature S and another signal X at slightly lower energies. S is located 1.81 eV above the VB maximum (VBM) of the ground state $S_0$, to which all following energies are referenced. At our incident laser fluence ($F \cong 0.2$ mJ cm$^{-2}$), the singlet signal amounts to roughly 1% of the VB signal, such that roughly one out of 100 molecules is excited by the pump pulses. The signal X ($E - E_{VBM} = 0.95$ eV) has roughly 20% more intensity than S (Extended Data Fig. 1d) and appears at lower energies. In 500 fs, the primary step is completed since S and X have disappeared and a third feature has emerged, which we attribute to the bitriplet $^1$TT ($E - E_{VBM} = 0.65$ eV) (note that the energetic position of the bitriplet with respect to the VBM is not the energy of the triplet exciton).

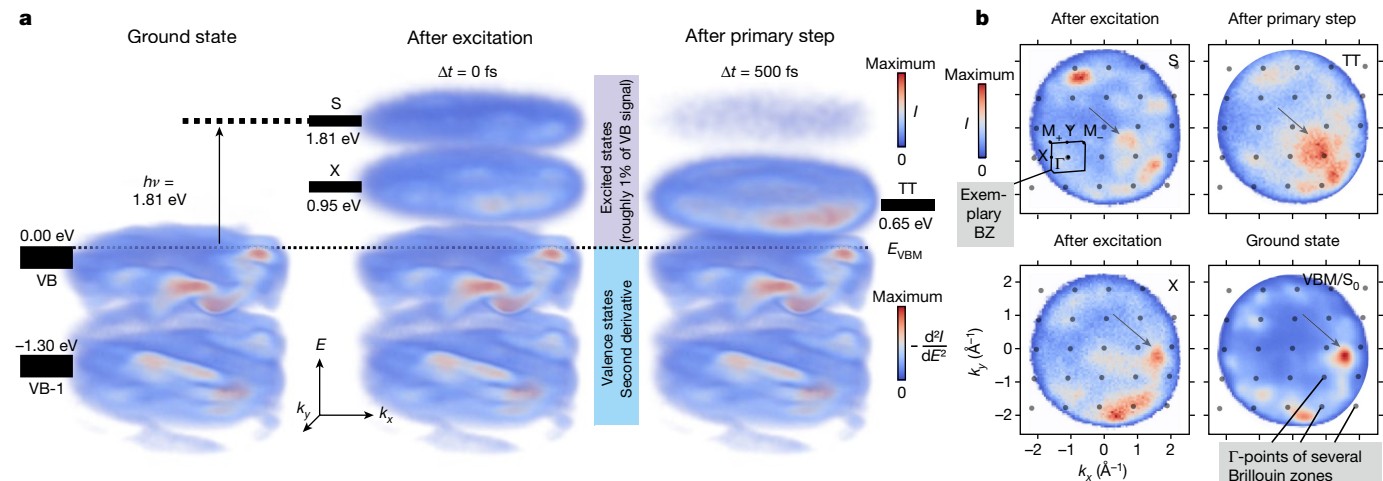

**Fig. 2 | Investigating SF with trARPES. a**, Visualization of the PE intensity of pentacene during SF. The energies of the states are referenced to the VBM. **b**, Momentum maps of the singlet exciton, the signal X (a satellite peak of the singlet exciton), the bitriplet and the ground state at the VBM. The arrows indicate high-intensity features related to the orbital character. For the momentum map of the ground state, the signal is shown at $E - E_{VBM} = 0.00$ eV; for those of the excited states, the signal was integrated over the following energy and time ranges: S (1.60 to 2.00 eV | −10 to 140 fs), X (0.95 to 1.30 eV | −10 to 35 fs) and TT (0.50 to 0.80 eV | 480 to 520 fs).

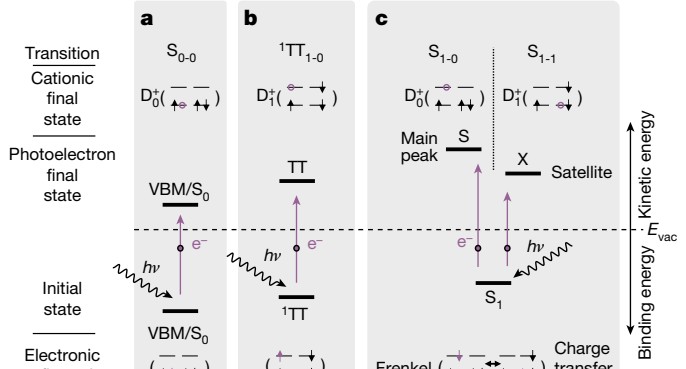

**Fig. 3 | PE transitions. a–c**, State diagrams and transitions of the relevant initial states, the ground state $S_0$ (**a**), the bitriplet exciton $^1TT$ (**b**) and the singlet exciton $S_1$ (**c**). Each transition yields a photoelectron and a cationic final state after PE.

Momentum maps of the states are shown in Fig. 2b. These seem rather disordered compared to those of inorganic crystals as a natural consequence of the low symmetry of the pentacene crystal and the slanted alignment of the molecules. In the momentum map S, representative of the singlet exciton, we distinguish three features. First, there are short-wavelength periodic peaks (spaced by roughly 0.9 Å$^{-1}$) of the PE intensity on the right side of the momentum map. Their periodicity is due to the large unit cell of pentacene. Second, these peaks are relatively sharp ($k_{FWHM} \approx 0.5$ Å$^{-1}$, Extended Data Fig. 2) as a consequence of the delocalization of the singlet exciton and hence its broad envelope function, which extends over several molecules[35,40]. Finally, there is a long-wavelength modulation of the PE intensity over the whole momentum map, causing it to peak at the top left and bottom right. This slow modulation derives from the lowest unoccupied molecular orbital (LUMO) character of the $S_1$ Wannier function[41].

In contrast to the singlet, the momentum map TT of the bitriplet exciton is less defined. The individual peaks are less pronounced ($k_{FWHM} \cong 1.0$ Å$^{-1}$) and there is no periodic pattern. By Fourier analogy, the bitriplet is highly localized in real space and the envelope function of the triplet is essentially confined to a single molecule[35,40]. However, there is noticeable similarity between the bitriplet and the singlet momentum map. Both feature the same long-wavelength modulation, peaking at the top left and bottom right. This modulation is due to their common LUMO character. The crucial difference between the two excitons is their degree of delocalization encoded in $F_{env}$.

The momentum map X looks substantially different from S and TT. It features pronounced peaks ($k_{FWHM} \cong 0.5$ Å$^{-1}$), but its PE intensity peaks at different locations in the momentum map. We attribute this to a different orbital character of the initial state of which X is a signature. We find that the momentum map of the VB maximum ($E = E_{VBM}$) closely resembles the shape of X. This indicates two things: (1) the electrons are similarly delocalized at the VBM and in the state from which X is derived and (2) both have a similar orbital character. As the VB of pentacene is predominantly formed by the highest-occupied molecular orbital (HOMO), X also has a HOMO character. The stark difference between the momentum maps of states with HOMO (X and VBM) and LUMO character (S and TT) reflects the different symmetries of the frontier orbitals at the surface. Having clarified the orbital character of X, we turn to the question of its origin.

To this end, we look at the highest-lying PE transitions relevant for SF (Fig. 3). Three different initial states are relevant, the VBM (Fig. 3a), the bitriplet (Fig. 3b) and the singlet (Fig. 3c). PE from the VBM ejects a photoelectron from a HOMO and leaves the electron ground state of the cation $D_0^+$ behind (HOMO to $D_0^+$). This is different for the bitriplet (LUMO to $D_1^+$), in which a triplet excitation remains in the system.

Last, there are two possible transitions for the singlet exciton, namely the main transition (LUMO to $D_0^+$) and the satellite (HOMO to $D_1^+$). The main transition probes both the Frenkel and CT character of $S_1$, whereas the satellite is a unique signature of the CT character. Because of the difference of the configurations of $D_0^+$ and $D_1^+$, the satellite should be located roughly a triplet energy $E_T = 0.86$ eV (ref. [42]) below the main peak. Because the signal X has HOMO character and is located $0.86 \pm 0.03$ eV below S, we assign it to the transition $S_{1-1}$. S and X are thus signatures of the same initial state $S_1$. This assignment has been proposed earlier[5], but neither a clear assignment by experiment nor by computations has been achieved.

With the knowledge about the involved states and their different momentum maps at hand, we look at the momentum-integrated dynamics of the excited states (Fig. 4a). We see the simultaneous population of S and a lower energy signal, as observed by Chan et al.[3]. There is a fast decay of S and an apparent energetic relaxation of the lower energy signal. The latter has led Chan et al. to the interpretation that this signal is the hot bitriplet, coherently formed by optical excitation, which quickly relaxes. Our knowledge about the orbital character of the transitions reveals the lower energy signal as a sum of two different, energetically overlapping transitions, namely $S_{1-1}$ and $^1TT$. The stark difference between the momentum maps of these transitions makes a decomposition of the signal in momentum space possible. This idea has been pioneered by Puschnig et al.[43] to deconvolute orbital contributions in ARPES data of ground state molecules. We extend this method to non-equilibrium states and use an experimentally measured reference, rather than computed momentum maps, as a basis set.

Consequently, we project the PE intensity onto a HOMO–LUMO basis, given by the momentum map X for a signal with HOMO character and TT for a signal with LUMO character as shown in Fig. 4b. From the momentum-integrated PE intensity, we then get an orbital-projected and time-resolved PE intensity as

$$I(E, \Delta t) = I_{HOMO}(E, \Delta t) + I_{LUMO}(E, \Delta t)$$
$$= [\alpha_{HOMO}(E, \Delta t) + \alpha_{LUMO}(E, \Delta t)] \times I(E, \Delta t),$$

where $\alpha$ characterizes the orbital populations of HOMO and LUMO obtained by a minimization procedure (Methods and Extended Data Fig. 3 and 7). In the HOMO contribution (Fig. 4c) the X is predominantly visible, whereas S and TT appear in the LUMO contribution (Fig. 4d). Hence, the decomposition separates energetically overlapping states on the basis of their orbital character intrinsically imprinted in momentum maps. We furthermore applied the decomposition procedure to a dataset in which the crystal was rotated by 10°. This yielded correspondingly modified momentum maps (Extended Data Fig. 4) and the decomposition resulted in the same dynamics. Similar dynamics result when integrating over the HOMO or LUMO characteristic features (Extended Data Fig. 5).

Having successfully disentangled the excited states, we turn to the kinetic analysis of SF. The orbital-resolved dynamics of the individual PE signals (Fig. 4e) demonstrate that S and X show the same dynamics (for non-normalized kinetics, see Extended Data Fig. 6a). On the other hand, the TT population rises after the optical excitation, concurrently with the decay of the singlet. This clearly establishes that $^1TT$ is not directly excited, but populated by the decay of $S_1$ through the primary step. At longer time delays, the population dynamics of $S_1$ feature a biexponential decay (Fig. 4f) with a substantially longer second decay time $t \cong 600$ fs. Simultaneously, the HOMO character vanishes in the momentum map of the lower energy signal (Fig. 4g). At longer delays up to 10 ps this momentum map does not change visibly.

We deem it likely that the origin of the 600 fs time constant is the separation of $^1TT$ to $^1T \cdots T$. In fact, the 2 ps timescale for bitriplet separation observed by Pensack et al.[24] in a pentacene derivative is similar to ours. The origin of the higher energy signal after 200 fs is then $^1TT$ and not $S_1$. Such a signal can be readily explained with a mixing of diabatic

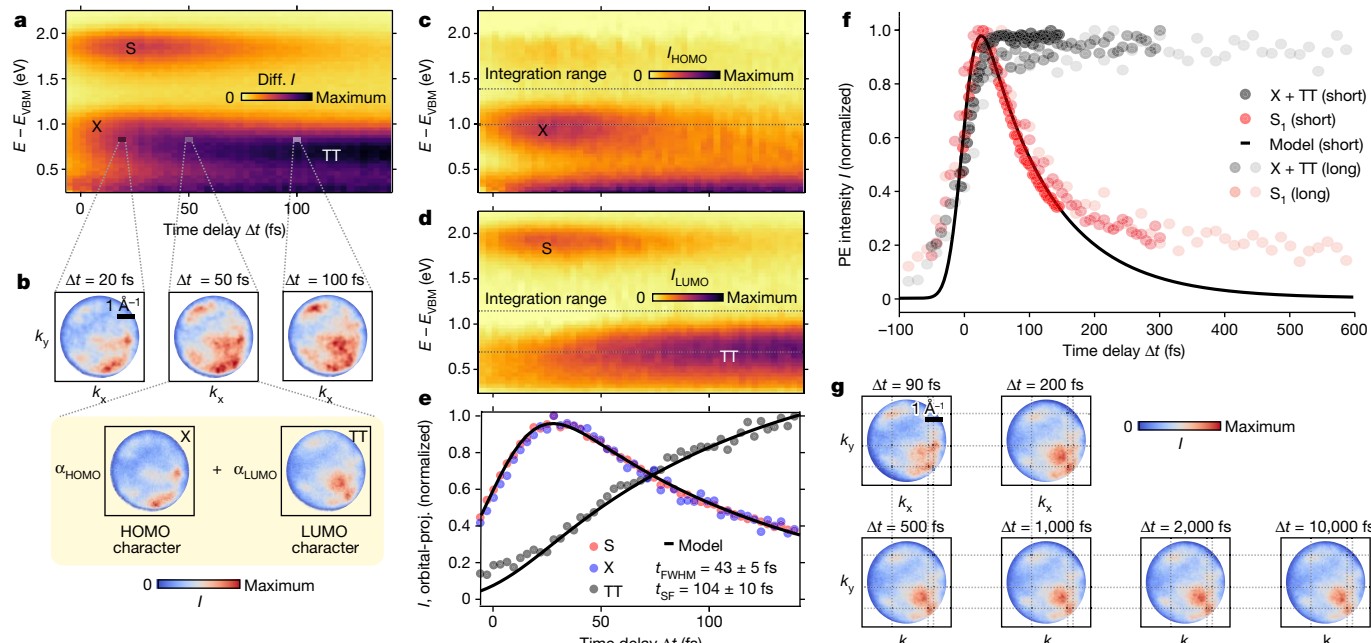

**Fig. 4 | Orbital-projected dynamics and evolution at longer delays.**
**a**, Momentum-integrated dynamics of SF in pentacene, showing the differential PE intensity (Diff. *I*, equilibrium signal subtracted). **b**, Momentum maps at $E - E_{VBM}$ and illustration of the decomposition procedure. **c,d**, The dynamics of states with HOMO (**c**) and LUMO (**d**) character. **e**, Orbital-projected (proj.) population dynamics of the excited states shown with the model fit. The signal was integrated over the shown energy range in **c** and **d** to reduce spurious counts from lower-lying states. **f**, Dynamics of the singlet and the lower energy signal at longer delays, not orbital projected. The short dataset and model fit are the same as in **e**. The long data sets were acquired with a slightly longer instrument response function. **g**, Momentum maps of the lower energy signal (integrated from 0.6 to 1.2 eV) at longer time delays.

bitriplet and CT states in the adiabatic bitriplet exciton (Extended Data Fig. 8). The CT character, which is much smaller than in the singlet exciton, is then lost in the transition to the non-interacting separated bitriplet exciton $^1T\cdots T$. Two more observations make our interpretation likely: (1) the higher energy signal shifts down in energy by >50 meV over the course of the primary step (Extended Data Fig. 9) and (2) the residual HOMO character that our decomposition procedure finds in the low-energy signal at 140 fs (Fig. 4c).

Our results let us return to the questions concerning the mechanism of the primary step raised at the beginning. In the coherent mechanism, $^1TT$ would be instantaneously populated. This stands in contrast to our observation. The bitriplet is formed after the optical excitation by the 100 fs decay of the singlet exciton. We can therefore safely rule out a coherent mechanism for SF in pentacene and hence the strong electronic or vibronic coupling between the bitriplet and the singlet excitons necessary to induce an instantaneous population.

Our observations show the nature of the electronic states and reveal the significant CT character of the singlet exciton. They do not show the nuclear rearrangement that follows the perturbation by optical excitation and SF. The nuclei will relax in both steps. Specific vibrational modes will be launched that shuttle the wave packet to lower energies. As such, our observations do not exclude a vibronic or a conical intersection mechanism based on high-frequency modes and are consistent with the observation of specific vibrational modes in the SF product states. At the same time, our observations do not provide any evidence for conical intersection dynamics, that is, as a modulation of the fission rate with the vibrational frequency. Mechanisms that do not rely on the assistance of specific vibrations but rather on the coupling to a phonon bath could be sufficient to explain SF dynamics in crystalline systems.

The direct evidence of physical mixing of CT states into $S_1$ is thus perfectly consistent with the purely electronic CT-mediated mechanism. Whereas the time constant for the primary step $t_{SF} = 270$ fs obtained by Berkelbach et al. is substantially larger than our value, the more recent many-body perturbation theory calculations in reciprocal space by Refaely-Abramson et al.[10] resulted in $t_{SF} = 30 – 70$ fs, in reasonable agreement with our observation.

Our analysis also offers a new perspective on SF in crystalline tetracene[44] and hexacene[45]. In the former, CT states lie energetically higher and CT mixing into $S_1$ is thus smaller. Nonetheless, the significant Davydov shift in tetracene indicates that in PE, the CT-induced transition $S_{1-1}$ should be visible. Hexacene features CT states that lie lower than in pentacene. We expect them to lie in between Frenkel and bitriplet states, leading to a stronger CT character of $S_1$ and a similar one of $^1TT$.

In conclusion, we have decomposed the excited states occurring during ultrafast SF in crystalline pentacene in momentum space by their orbital character. The orbital resolution allowed us to infer a large CT character of the bright singlet exciton and rule out the coherent mechanism with an instantaneous population of the bitriplet exciton. Our analysis proves that the depth of information contained in trARPES data allows disentangling the dynamics of ultrafast molecular processes: excited states can be unambiguously assigned to and separated by their orbital character imprinted in momentum maps. We believe that the presented orbital-resolved analysis will have ramifications for a range of molecular processes that have not been decoded yet. Furthermore, our state-resolved approach should grant access to a deeper view into the ultrafast dynamics of topological matter, governed by the underlying transient wave functions.

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

# Article

## Methods

### trARPES

We used a homebuilt high-harmonic generation setup operating at 500 kHz to provide XUV pulses for PE[38] with the following parameters: photon flux was up to $2 \times 10^{11}$ photons per second on target, spectral width was 110 meV full-width at half-maximum (FWHM), photon energy was 21.7 eV and pulse duration 35 fs. As the visible (VIS) pump beam, we used the compressed output of a non-colinear optical parametric amplifier with the following parameters: incident fluence of roughly 0.2 mJ cm$^{-2}$ and photon energy of 1.81 eV. The instrument response function of the main dataset (Fig. 4e) was 43(5) fs, obtained by fitting the kinetic model mentioned in the main text. A longer instrument response function ($60 \pm 15$ fs) is obtained for the long data sets shown in Fig. 4f. These data sets were acquired under slightly different experimental conditions. Both VIS and XUV pulses were incident in parallel at 65° with respect to the surface normal. The VIS pulses were s-polarized along the $a$ axis of the crystal and the XUV pulses p-polarized. Under these conditions, we estimate that roughly 2.5% of molecules are in an excited state. The pentacene crystal was glued with conductive ultra-high vacuum glue to a copper sample holder and then mounted on a six-axis cryogenic manipulator (SPECS GmbH) and cleaved at a base pressure of $5 \times 10^{-11}$ mbar. The data were acquired at room temperature using a time-of-flight momentum microscope (METIS1000, SPECS GmbH), allowing detection of each photoelectron as a single event and as a function of pump-probe delay. The resulting 4D PE intensity data is, hence, a function of the two components of the parallel momentum, the electron kinetic energy and the pump-probe time delay: $I(k_x, k_y, E, \Delta t)$. The overall energy and momentum resolutions of the experiment are $\Delta E = 150$ meV and $\Delta k = 0.08$ Å$^{-1}$, respectively[46]. Owing to the damage that the XUV radiation causes in pentacene, we maximized the field of view (field aperture diameter of 500 μm) and therefore the number of probed molecules, while still maintaining the momentum resolution at the relevant kinetic energies. We constantly operated at the count rate of roughly $1 \times 10^6$ cts s$^{-1}$ to acquire the data sets in reasonable time. The two main data sets used in this publication have been acquired over a period of roughly 30 h, respectively. The first dataset, used in Figs. 2a and 4g, was acquired in a 'snapshot' mode, in which the time delay was not continuously changed but fixed to certain values. The second dataset, used in all other figures, was acquired in a 'continuous' mode, in which the time delay was continuously changed. The extra data sets shown in Fig. 4f were acquired over less than 1 h.

### Signal decomposition

To decompose the dynamics of the two transitions $S_{1-1}$ and $^1$TT, we use the momentum information we have for each energy-time pixel and the fact that the excited states signal is composed of transitions with either HOMO or LUMO character. We can hence project the momentum map at a given energy-time pixel $I_{E_0, \Delta t_0}(\mathbf{k})$ to a HOMO−LUMO basis. The basis consists of the momentum map of X, $\mathbf{X}(\mathbf{k})$, for a signal with HOMO character and that of $^1$TT, $\mathbf{TT}(\mathbf{k})$, for a signal with LUMO character as shown in Fig. 2b. Before the projection, we normalize all momentum maps as $\frac{I}{\int I dk_x dk_y}$ and convolute the 4D PE intensity with a Gaussian ($\sigma_k = 0.05$ Å$^{-1}$, $\sigma_E = 40$ meV, $\sigma_t = 3$ fs) making the procedure more stable. The projection then allows us to extract the coefficients $\alpha_{HOMO}$ and $\alpha_{LUMO}$ for each energy-time pixel ($E_0, \Delta t_0$) by minimizing.

$$\chi^2 = \int (I(\mathbf{k}) - \alpha_{HOMO}\mathbf{X}(\mathbf{k}) - \alpha_{LUMO}\mathbf{TT}(\mathbf{k}))^2 dk_x dk_y,$$

where $\alpha_{LUMO} = 1 - \alpha_{HOMO}$. The extracted coefficients are shown in Extended Data Fig. 3. Regarding the quality of the decomposition, it is essential to have a well-defined set of basis maps, with one transition being dominant in each map. In our case, one basis is formed by the bitriplet after a sufficient time delay, such that there is no mixing with residual singlet satellite. The other basis map is the momentum map X, integrated over an energy-time window wherein the singlet satellite is dominant (Fig. 2b). Owing to the time resolution of the experiment, a small amount of residual bitriplet (<10% of total signal) remains mixed into this momentum map. Some ambiguities in the decomposition hence remain, such as the artefactual signal in the HOMO-projected density of states at 1.8 eV. We further validated the decomposition procedure by investigating the dynamics of features specific to HOMO and/or LUMO (Extended Data Fig. 5).

### Kinetic model

The model used in the main text is a Markovian model with two components and the rate constant $k_{SF}$. The model results in the system of differential equations

$$\frac{d[S_1]}{dt} = -k_{SF}[S_1] + I_{SRF}$$
$$\frac{d[^1TT]}{dt} = k_{SF}[S_1] \tag{1}$$

where $[S_1]$ and $[^1TT]$ are the populations of singlet and bitriplet. $I_{SRF} = e^{\frac{-t^2}{2\sigma}}$ is the system response function. This model was fitted to the singlet and the satellite signal, which yields the rate constant and the width of the system response function. The errors given in Fig. 4e are due to the systematic uncertainty of the decomposition remnant in the satellite signal. The error of the fit to only the singlet signal is much smaller and in the range of 2 fs.

### Sample degradation

Just like many other organic materials, pentacene experiences light-induced damage when irradiated by XUV light. Throughout the measurement of one full dataset, the VB intensity reduces by 25%, indicating a chemical modification. This leads to a slight blur of the momentum map at the end of the measurement. However, the overall shape and features of the momentum maps are conserved (Extended Data Fig. 6b−d). Whereas it would be, in principle, desirable to reduce the degradation, we note that there is a trade-off between acquiring enough signal in the excited states requiring a long measurement time and reduced degradation requiring a short measurement time (Extended Data Fig. 7). We optimized these parameters and further reduced the XUV flux experienced by the pentacene sample by enlarging the field aperture of the momentum microscope.

### Materials

Pentacene single crystals were grown by means of horizontal physical vapour deposition[47,48]. About 50 mg of pentacene (purified by two-fold gradient sublimation) is placed in a horizontal furnace. By simultaneously applying a sharp temperature gradient and a 30 sccm N$_2$ (6N purity) transport gas flow across the furnace, the pentacene is sublimed at 290 °C and then transported along the temperature gradient towards the cold side. There, defined, plate-like single crystals grow by recrystallization of supersaturated pentacene vapour. The 96 h growth period finishes with an extra 12 h cool-down phase to minimize thermal stress. As analysed elsewhere[49], the crystals adopt the low-temperature bulk phase[50,51] and the plate-like habit is dominated by the large (001)-facet that was consequently probed by our trARPES measurements. Thus, we can investigate the dynamic population of the excitonic band structure between the photophysically relevant nearest neighbours within the pentacene crystal.

### Data availability

The datasets generated during and/or analysed during the current study are available in the Zenodo repository, https://doi.org/10.5281/zenodo.6451829.

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

**Acknowledgements** We thank H. Seiler for helpful discussions. We gratefully acknowledge funding from the Max-Planck-Gesellschaft, the Deutsche Forschungsgemeinschaft within the Emmy Noether program (grant no. RE 3977/1) and through SFB951 (grant no. 1820877777, project B17) and the priority program SPP2244 (project 443366970), the Max Planck-EPFL Center for Molecular Nanoscience and Technology and the European Research Council (ERC) under the European Union's Horizon 2020 research and innovation program (grant no. ERC-2015-CoG-682843 and grant agreement No 899794). S.B. acknowledges financial support from the NSERC-Banting Postdoctoral Fellowship Program. T.P. acknowledges funding from the Alexander von Humboldt Foundation. S.H. and J.P. acknowledge financial support by the Bavarian State Ministry of Science, Research, and the Arts (Collaborative Research Network 'Solar Technologies Go Hybrid'). S.H. gratefully acknowledges funding from the German Research Foundation (DFG) through the project no. 490894053.

**Author contributions** A.N., S.B. and R.E. conceived the research project. S.H. and J.P. synthesized the sample. S.D. built the spectrally tuneable pump beamline with help of S.B. A.N., S.B., S.D., T.P., J.M., R.P.X. and L.R. performed the experiment. A.N. acquired and analysed the data. R.P.X. wrote and maintained the code to analyse the data. A.N. wrote the original draft with significant input from S.B., S.H., S.D. and J.M. All authors discussed the results and reviewed the manuscript. M.W., L.R., J.P. and R.E. provided the project infrastructure and funding.

**Funding** Open access funding provided by Max Planck Society.

**Competing interests** The authors declare no competing interests.

**Additional information**
**Correspondence and requests for materials** should be addressed to Alexander Neef or Ralph Ernstorfer.

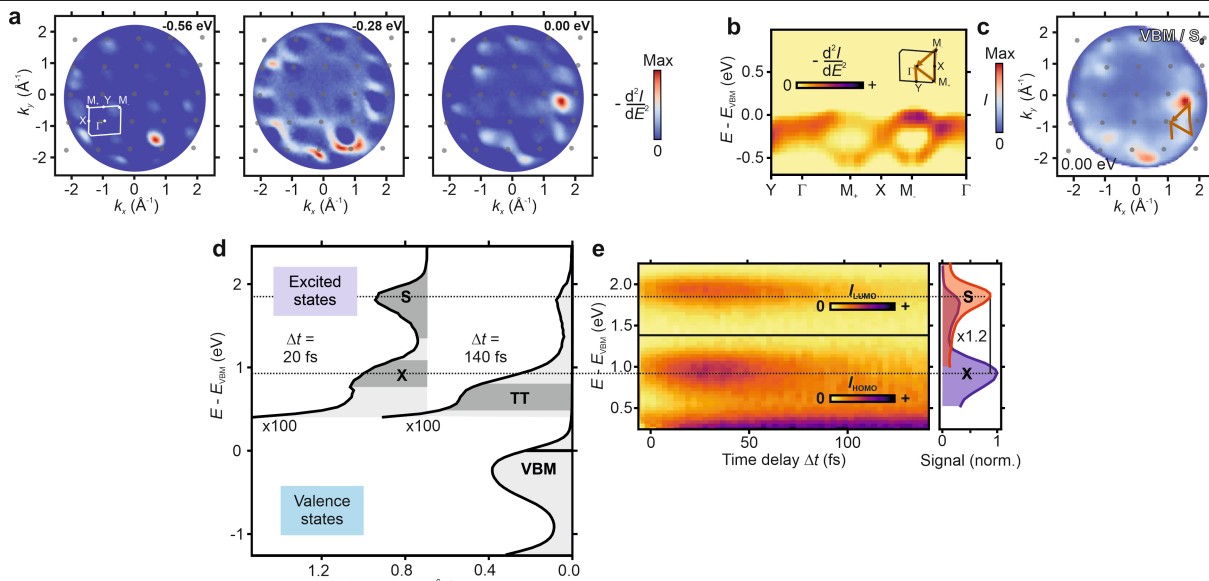

**Extended Data Fig. 1 | Reciprocal lattice, band structure and momentum-integrated spectra. a**, Momentum maps of the second derivative of the valence bands at three different energies, corresponding to its minimum, center and maximum. The clearly visible extremal points, especially at the VB minimum, make it possible to align the reciprocal lattice with the data. **b**, The second derivative of the PE intensity along the given path in momentum space. At the M-points the band separation peaks at 500 ± 20 meV, which is extracted by Gaussian fits to the energy distribution curve. **c**, The momentum path of **b** is overlayed on the momentum map of the VBM. **d**, PE intensity at two different delays. **e**, Orbital-projected PE intensity of the singlet exciton. On the right the signal is integrated from -10 fs to 20 fs.

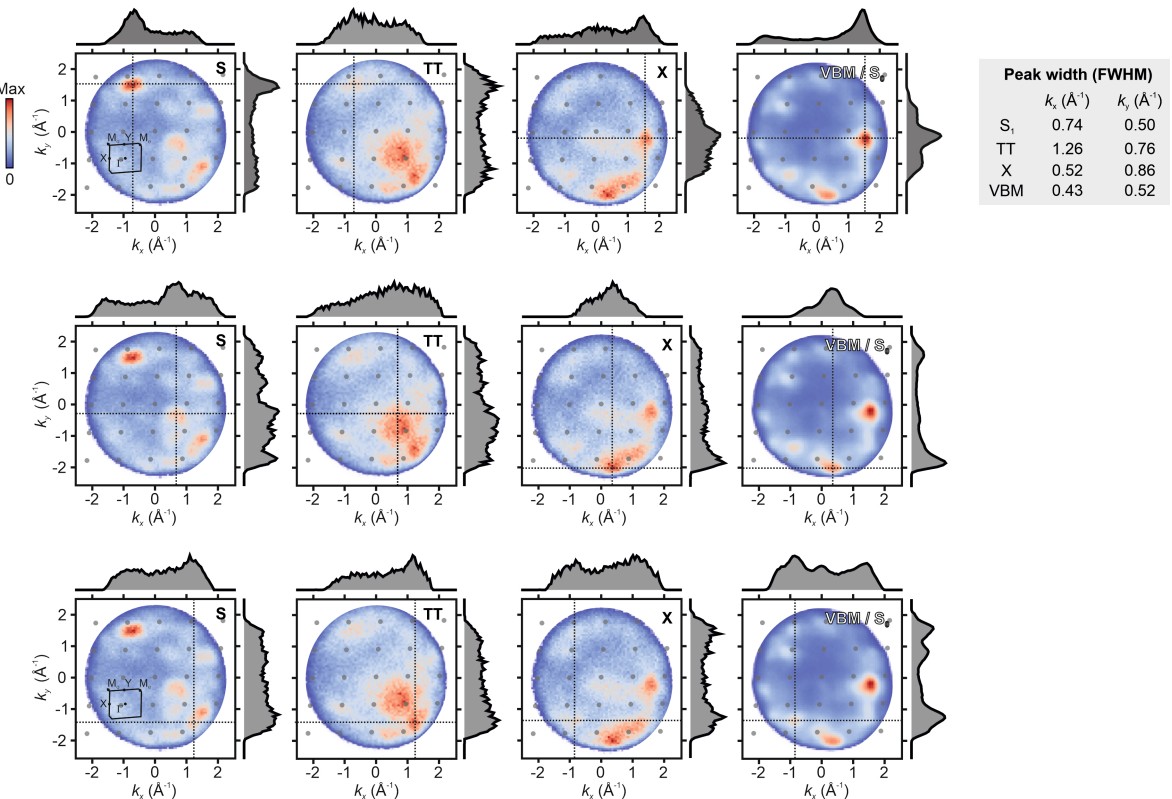

| Peak width (FWHM) | | |
|---|---|---|
| | $k_x$ (Å$^{-1}$) | $k_y$ (Å$^{-1}$) |
| S$_1$ | 0.74 | 0.50 |
| TT | 1.26 | 0.76 |
| X | 0.52 | 0.86 |
| VBM | 0.43 | 0.52 |

**Extended Data Fig. 2 | Momentum maps and peak widths.** Momentum maps S, TT, X and VBM. The momentum distribution curves (MDC) along the respective cuts in momentum space are shown for each state. The full width at half maximum (FWHM) of the MDCs is evaluated for the indicated peak in the top row.

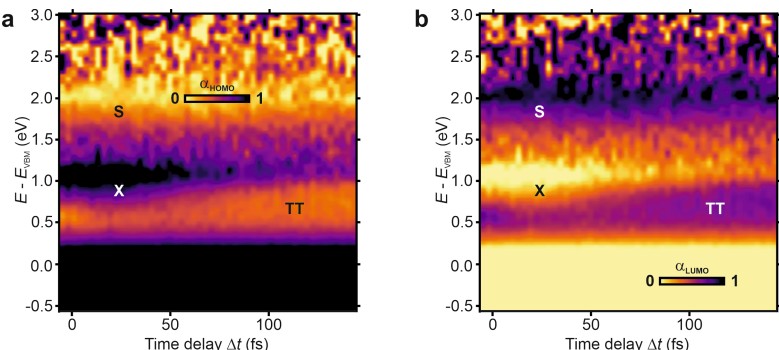

**Extended Data Fig. 3 | Orbital populations.** Orbital populations as obtained by the minimization procedure of states with **a**, HOMO and **b**, LUMO character.

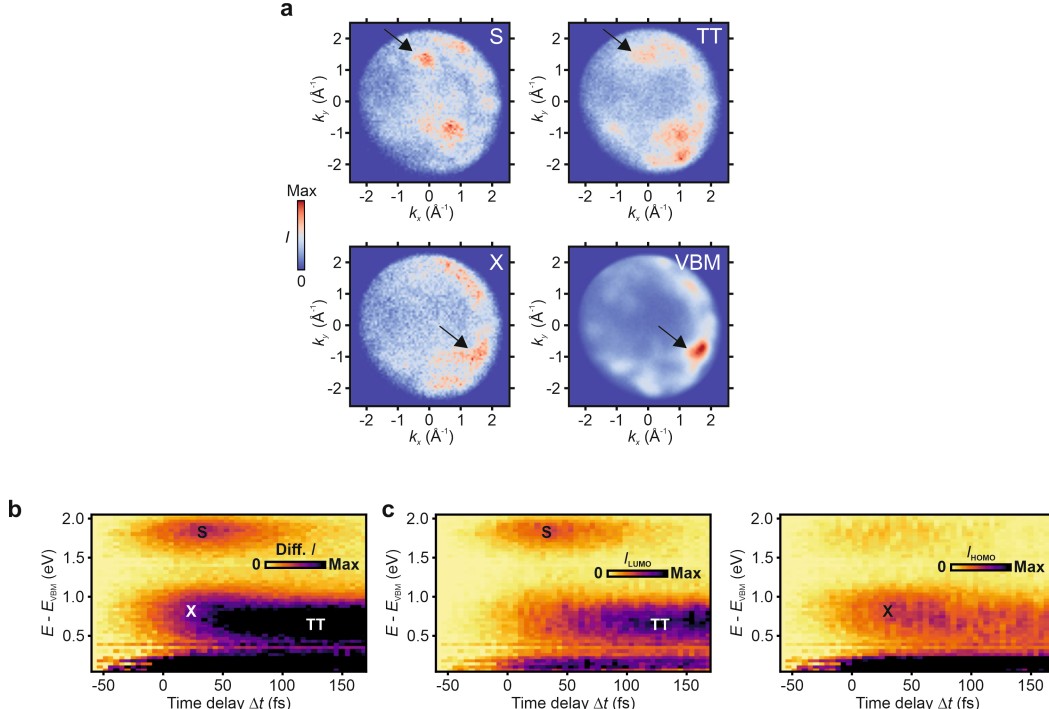

**Extended Data Fig. 4 | Momentum maps and projected dynamics for the crystal rotated 10° about the surface normal. a**, Momentum maps of the singlet exciton, the triplet exciton, the satellite X and the valence band maximum. For the momentum map of the ground state the signal is shown at $E - E_{VBM} = 0.00$ eV, for those of the excited states, the signal was integrated over the following energy and time ranges: S: (1.60 eV to 2.00 eV | -10 fs to 140 fs), X: (0.95 eV to 1.30 eV | -20 fs to 60 fs), TT: (0.50 eV to 0.80 eV | 580 fs to 620 fs). **b**, Momentum-integrated dynamics, equilibrium signal substracted. **c**, Orbital-projected dynamics with the maps T for LUMO character and X for HOMO character.

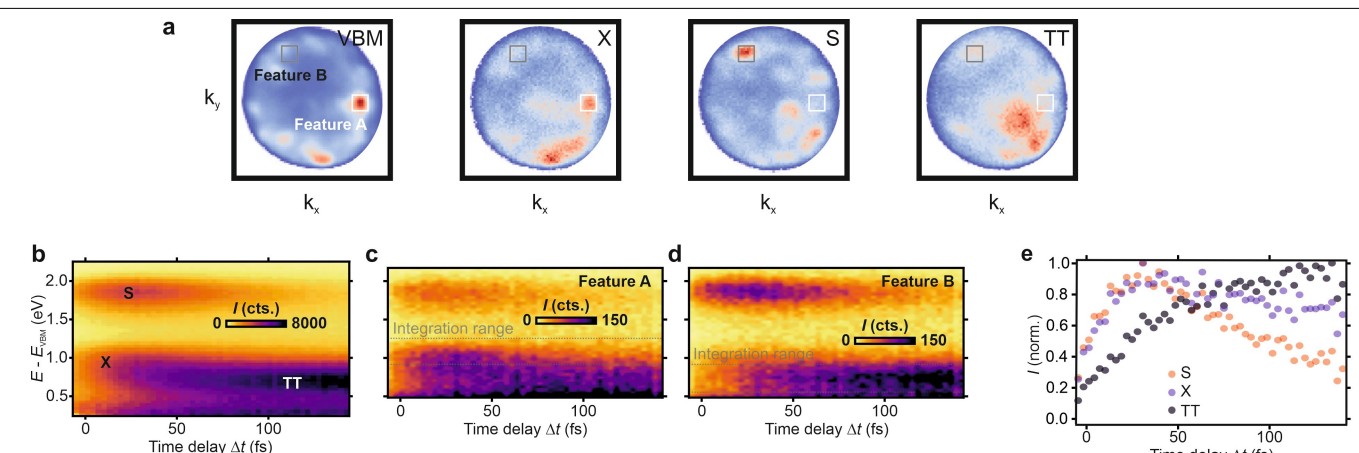

**Extended Data Fig. 5 | Dynamics of HOMO and LUMO features. a**, Selected features characteristic of HOMO and LUMO character on the momentum maps of VBM, X, S, and TT (the same as in Fig. 2 of the main text). **b**, Dynamics of signal integrated over all momenta, shown here for comparison. **c** and **d**, Dynamics of signal integrated over the squares around the characteristic features A and B. **e**, Energy-integrated dynamics of S, X and TT. The signal was integrated in the shown range.

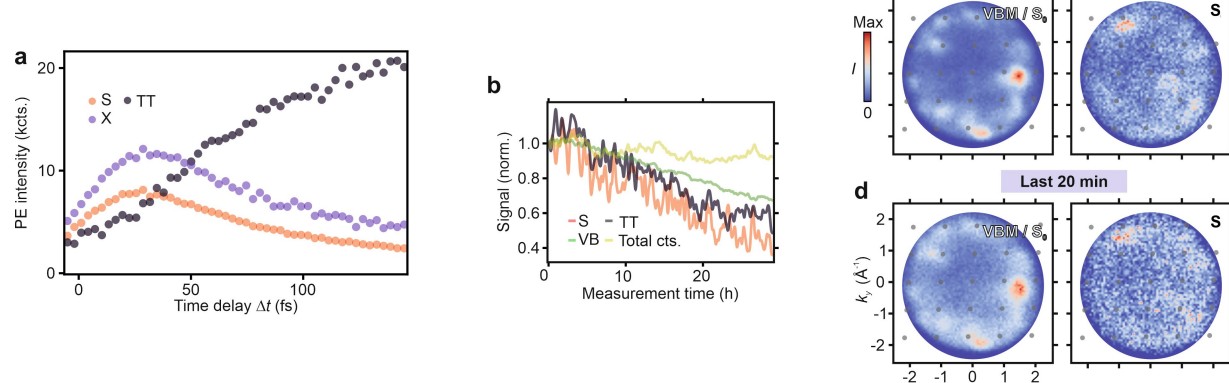

**Extended Data Fig. 6 | Non-normalized dynamics and sample degradation.** **a**, Non-normalized dynamics of orbital-projected S, X and TT. **b**, Degradation of the signal of S, TT and VBM over the course of a measurement. For comparison the total count rate, i.e. all measured electrons, is shown. **c**, Momentum maps of VBM and S of the first 20 minutes of measurement compared to **d**, the last 20 minutes of the same data set as in **b**.

| Measurement time | Total PE counts in VB | Total PE counts in Singlet | LUMO-projected | HOMO-projected |
|---|---|---|---|---|
| 25 min | 7.5 Mcts. | 15 kcts. | | |
| 75 min | 23 Mcts. | 44 kcts. | | |
| 4 h | 80 Mcts. | 150 kcts. | | |
| 11 h | 220 Mcts. | 420 kcts. | | |
| 27 h | 480 Mcts. | 730 kcts. | | |

**Extended Data Fig. 7 | Statistics and decomposition.** Orbital-projected dynamics for a range of measurement times and respective total counts in the VB and the singlet exciton. All plots are based on the main dataset and normalized to the total count rate in the VB for each row.

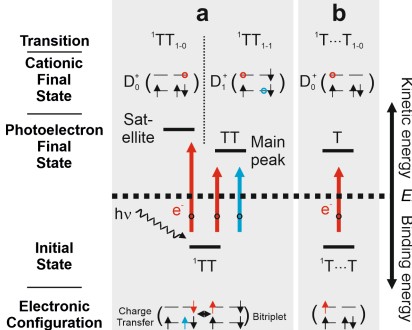

**Extended Data Fig. 8 | Additional photoemission transitions.** State diagram of the different initial states and the photoelectron and cationic final states after photoemission. **a**, Bitriplet exciton with charge-transfer character and **b**, separated, non-interacting bitriplet exciton. Transitions from the LUMO are shown in red, transitions from the HOMO in blue.

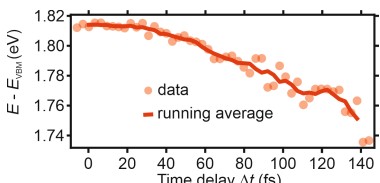

**Extended Data Fig. 9 | Position of the higher energy signal.** Time-dependence of the energetic position of the higher energy signal of the main high-statistics dataset. The energetic position was obtained by Gaussian fits to the momentum-integrated signal.