## [Peer Review File · Nature]

Manuscript Title: Orbital-resolved observation of singlet fission

Reviewer Comments & Author Rebuttals

Reviewer Reports on the Initial Version:

Referee #1 (Remarks to the Author):

The authors report the hitherto most direct and clearest experimental evidence for the mechanism of singlet fission in the model system of crystalline pentacene. Using tr-ARPES with unprecedented momentum mapping applied to the singlet fission problem, the authors provide unambiguous delineation of the excited states from their frontier molecular orbital (MO) characters. In particular, the authors resolved a long standing puzzle on the initial step in singlet fission on the ultrafast time scale, namely, how is the $1(TT)$ state formed from the optically bright $S1$. The MO characters in the momentum map allow the authors to conclude that the $S1$ is of partially CT character, an idea long proposed in the singlet fission literature, but never before so clearly established. This provides a re-interpretation of the near instantaneous formation of $1(TT)$ observed by Chan et al. (ref. 3). The photoionization of the CT state yields HOMO character in the momentum maps, confirming a hypothesis in the Supporting Info of Yost et al (ref. 5) on the interpretation of earlier tr-2PPE signal by Chan et al. Moreover, the finding suggests the dominance of electronic coupling, at least in the pentacene system, and argues against another proposal of conical intersection (ref. 6 & 7). I congratulate the authors for the success in carrying out this brilliant experiment. This is a major milestone in experimental research on singlet fission, since the renaissance of this field 12 years ago. This work should serve to guide future mechanistic research on singlet fission. I strongly support the publication of this work in Nature.

The following are optional comments for the authors to consider.

- 1) The authors cited Berkelbach et al's seminal paper on the proposal of CT mediated singlet fission. One thing the authors may want to clarify is that Berkelbach's proposal is mainly on a super exchange type of involvement of CT states, not the actual population of the CT state as is observed experimentally here. There are a number of papers on the role of actually populated CT states in singlet fission, as summarized in ref. 17. Also, the authors cited ref. 32 in the discussion of orbital characters. Ref. 32 also showed the major CT character of $S1$.
- 2) In the present system of crystalline pentacene, the $S1$ state is of predominantly CT character. The authors may want to discuss how general this mechanism is when CT is energetically not resonant with FR. Model systems such as tetracene or hexacene come to mind.
- 3) While the coherent mechanism may not be dominant here, the authors may want to consider the following. For an ultrafast process happening on the 70 fs time scale, it is difficult to distinguish dephasing from incoherent population decay. For the momentum map of the X state right after photo-excitation, one can also see features of the $1(TT)$ state. The authors decomposition procedure clearly yields non-zero intensity for the $1(TT)$ state at time zero. This suggests $\sim 20\%$ coherent formation of $1(TT)$. After all, if there is sufficient electronic coupling, quantum mechanics teaches us that the states are mixed within the spirit of uncertainty principle, provided dephasing rate is not overwhelming. The authors may want to add some clarification to this point.

Referee #2 (Remarks to the Author):

In the article 'Orbital resolved observation of singlet fission', authors Alexander Neef et al have used time-resolved momentum microscopy to map out the orbital character of the transient state in the process of exciton fission in crystalline pentacene. Thereby, they identify the process of fission as being due to a charge-transfer mediated mechanism, an issue that was still debated in literature. Utilizing the powerful knowledge provided by orbital-resolved measurements, Neef et al seek to provide insightful understanding into a process that is important to various aspects of material science, particularly to photovoltaic efficiency since the singlet fission process has the potential to double the number of excited charge carriers.

Overall, I find this type of study innovative and powerful. I think the additional information that orbital resolved techniques bring to the table, give a nice holistic picture of the process. The authors have clearly been able to go beyond the information obtained so far, which largely provided only the energetic positions of the transient states. I find the manuscript well written. However, I have some major concerns (and some minor concerns/suggestions) that need to be addressed before the manuscript is reasonable for publication:

1. I am quite confused by the heart of the data – the momentum maps in fig. 2b. I would have expected some sort of symmetry/repetitive structures to this measurement corresponding to the many Brillouin Zones that are measured. I am surprised that the momentum maps look so 'disordered' in comparing neighboring BZs. My primary concern is that there are experimental artefacts associated with the alignment of the momentum microscope. My alternate concern is that photoemission matrix elements obfuscate the critical physics that they authors are claiming to observe.
 - a. For the first, can the authors provide convincing data that the transmission function through the microscope of the photoemitted electron from sample to detector is not the primary cause of the measured distribution? Typically, aberrations and imperfections in alignment lead to a transmission probability that depends on k and E . Such errors could severely impact the measured distributions in Fig.2b and also impact the conclusions.
 - b. What kind of distribution does one expect from a theoretical calculation? It is not too difficult to calculate the photoemission intensity map $I(k, E)$ for the S state and for the HOMO and LUMO (at least). One can also include photoemission matrix elements in such a calculation. Are the authors able to theoretically reproduce the distribution they measure? From the experimental perspective as well, one can rotate the sample or the probe polarization to reduce or symmetrize the effects of the photoemission matrix elements. Can the authors experimentally confirm this?

2. Organic samples are notorious for being damaged during the photoemission process. It would be important to confirm experimentally that the sample isn't being damaged during the process of measurement for the given measurement conditions. Damage to the sample during the measurement process could also be an explanation for the 'disordered' momentum maps. To confirm no damage to the sample, one could continuously measure the sample over a period much longer than the longest experimental window reported in the paper. Then one could confirm that the data accumulated in the first fraction of this measurement matches the data obtained in the last

fraction of the same measurement.

(Minor Comments)

3. The authors introduce the phenomenological envelope function $F_{env}(R)$ after discussing the initial wavefunction as a linear combination of Wannier functions w . However, I wasn't able to get the quantitative connection between the two quantities. Is there a way to present the envelope function $F_{env}(R)$ quantitatively?

4. The authors describe that at a fluence of 0.2 mJ/cm^2 , the singlet provides about 1% of the VB signal. How does this compare to the known absorption in the material at the pump wavelength?

5. In the manuscript, the authors excite at 1.81 eV with polarization along the a -axis of the crystal. It would be helpful to give the context of the measurement, e.g. in order to excite the S state resonantly.

6. Related to the above question, can the author also comment the dynamics that allow a significant occupation of the X state after resonant excitation to the S state within the $\sim 30 \text{ fs}$ temporal resolution of the experiment?

7. As a suggestion, I found the 3D figures in 2A hard to understand. I leave it to the authors to see if this can be improved.

8. In Fig. 4e and f, it would be helpful to see the non-normalized datasets. Also, why does the data for TT terminate at 150 fs ? What happens after 150 fs now that the state is populated?

Referee #3:

The manuscript by Neef et al presents advanced time-resolved photoemission spectroscopy study of early singlet fission dynamics in pentacene. The study and results are following closely XYZhu study from 2011 (ref 3), but, in addition to the states' energies, authors record the momentum maps to recover additional information about the states' localisation. Experimental data are unique and well-presented. The timescales of fission and the respective energies of the states agree well with previous studies on pentacene. The key novelty claim by the authors is the observation of CT character in the state 'X' preceding the bi-triplet formation.

My key concern with this (experimentally very sophisticated and very well-performed) work is that no strong evidence is provided for CT character of state X. A number of qualitative speculations are put forward, however, no quantitative analysis is given to evaluate the likeliness and the extend of CT contribution. My (may be rather naïve) view is that momentum maps for state X can actually be seen as a superposition of momentum maps of S and TT. This would indicate that X is a mixed S-TT 'hot' state, may be the same as one observed in the analysis of coherent vibrionic dynamics. Such interpretation would not require CT state involvement, except via superexchange mechanism.

I also have a number of more specific comments:

1) figure 2a - the choice of presentation of momentum maps is artistic, but i feel use of 3D surfaces complicates interpretation.

2) time resolution of the experiment (sadly comparable to fission timescale) is not commented on in the main text, only in SI. I believe it is an important limitation of the method which should be discussed.

3) 1% of excited molecules corresponds to a very high pump fluence. Do authors see any power dependence, particularly after 100 fs where bimolecular process can kick in?

4) I find the separation of fission dynamics into three steps just based on the exponential timescales speculative. Are there evidence for clear systematic changes and energies or/and momentum maps on the timescales of 300 and 600fs?

5) the energy of state X is exactly half of the singlet energy - is it a pure coincidence? Can this point towards the validity of 'simple' mixed hot S-TT state interpretation?

Overall, I feel this is experimentally very strong study, however the interpretation of observing CT contribution is not based on solid evidence. May be some alternative explanation can brought forward and discussed. Taking the innovative character of the work but the limited new photophysical insight, it would be well suited for Nature Communications after the above comments are addressed

Referee #4 (Remarks to the Author):

Neef et al report trARPES measurements on one of the archetypal singlet fission systems, crystalline pentacene. The mechanism of ultrafast singlet fission in this system has been grounds for extensive debate since 2010, with the most prominent models invoking electronic coherence, vibronic effects/conical intersections, and mediation by coupling to charge-transfer states (typically, virtual CT state in a superexchange pathway). The CT mechanism was posed against models invoking direct coupling between the Frenkel-exciton S1 state and TT, which were latterly discarded as being unrealistic in the solid. On the experimental side, following the surprising report of coherent fission with an instantaneously populated TT state, later work coalesced around the vibronic models with interpretations grounded in the CT-mediated mechanism proposed by theory. Importantly, the optical spectra used to assign states in these earlier studies and the lack of control over CT states in films meant that it has never been possible to directly experimentally verify the involvement of CT in the solid state (in contrast to intramolecular singlet fission, where the model of CT mediation has been indirectly and directly verified, e.g. Busby, Nat Mater 2015; Margulies, Nat Chem 2016; Alvertis, JACS 2019). Nor have any subsequent models been able to directly explain the signatures assigned to coherent fission in 2011. In this context, the authors' new approach of utilizing momentum- and energy-resolved data to decompose the signals on the relevant ultrafast timescale presents a powerful advantage. According to the authors' analysis, the technique permits direct identification of the initial photoexcited state as a mixture of Frenkel singlet and charge-transfer excitations, as previously determined by steady-state spectroscopy (Yamagata, J Chem Phys 2011).

This coherent mixed state evolves directly into TT on the timescale known from optical spectroscopy. To the extent the analysis is robust (see below), this result reconciles the early 'coherent' observations with current theories and definitively validates the CT-mediated mechanism. It is sure to be of great interest to the singlet fission community and highlights the great potential of trARPES to understand other complex ultrafast processes.

However, much of the analysis hinges on the claim that the signals can be cleanly decomposed into HOMO-like versus LUMO-like contributions, allowing a decomposition into the X versus S and TT states. From the data provided, it's not clear this is fully justified. In the 1TT momentum map (Fig 2b), the authors highlight the feature in the upper left as characteristic. But stronger features are found in the lower right, including three detectable peaks that appear at precisely the same momenta in the X map. In Fig 4b, I struggle to see any features in the X map that are not present in the TT map, so how can they be cleanly decomposed? What is the meaning of X-like/LUMO-like features in the TT momentum distribution? Is it an oversimplification to project the detected momentum distributions onto these pure diabatic states? What would be the consequence of mixed states (for instance, 1TT is widely held to exhibit both CT and S1 wavefunction character)? As described in the text, this method of separating contributions based on energy and momenta is compelling, but the details cause some concern.

In this regard, the authors claim that "the crucial difference between the [S1 and TT] state is their degree of delocalization", which implies the principal difference in their momentum maps should be the linewidth. Yet there is a strikingly different intensity distribution as well, with many additional peaks in the bottom and right of the TT map, suggesting other differences in the orbital character that are not addressed or explained by the schematic in Figure 3.

In the conclusions, the authors nicely explain how their observations rule out the original coherent model. But it is not clear how the argumentation in lines 175-184 relates to the observations in the present work, or on what basis it rules out the involvement of a vibronic mechanism. The model is agnostic to the nature of the initial singlet state, and it remains entirely compatible with the kind of CT mediation invoked here (indeed, some of the same authors explicitly invoke CT-mediated coupling within the SI scheme in subsequent work, Schnedermann et al., Nat Commun 2019). If the authors are arguing that the vibronic mechanism is incorrect, can the present model account for the presence of coherent vibrations reproducibly observed in the SF product state TT? It seems that the present study demonstrates a powerful tool to understand the nature of the states involved in the singlet fission pathway – an important advance – but not the relevant couplings that drive the process. Momentum resolution yields a more detailed assignment of states, but the end result is still a kinetic model of the form $A \leftrightarrow B \rightarrow C$. The rate constants are purely empirical and it still falls to theory to map these onto matrix elements.

In addition to addressing these points, the authors should update their framing of the CT-mediated model of singlet fission to note it has been substantially validated in intramolecular systems.

Minor points:

-I believe the ratio of [TT]/[S1] should be 5, rather than 0.2.

-I could find no discussion of statistics or error bars. While the data looks clear, there should be a

discussion of signal vs background counts and noise levels.

-In extended data Fig 3, what is the justification for the choice of peaks used for linewidth analysis?
In the TT and X maps, other and stronger peaks appear to be available, with what seem by eye to be different widths. How representative/meaningful is the parameter currently extracted?

Author Rebuttals to Initial Comments:

Reply to referees' comments

Throughout this reply comments by the referees are marked in grey Courier font and answers by the authors in blue. Quotes from the revised manuscript are indented, *italic* and in black.

Reply to Referee #1 (Pentacene solar cells and singlet fission)

Referee #1: The authors report the hitherto most direct and clearest experimental evidence for the mechanism of singlet fission in the model system of crystalline pentacene. Using tr-ARPES with unprecedented momentum mapping applied to the singlet fission problem, the authors provide unambiguous delineation of the excited states from their frontier molecular orbital (MO) characters. In particular, the authors resolved a long standing puzzle on the initial step in singlet fission on the ultrafast time scale, namely, how is the $1(TT)$ state formed from the optically bright S_1 . The MO characters in the momentum map allow the authors to conclude that the S_1 is of partially CT character, an idea long proposed in the singlet fission literature, but never before so clearly established. This provides a re-interpretation of the near instantaneous formation of $1(TT)$ observed by Chan et al. (ref. 3). The photoionization of the CT state yields HOMO character in the momentum maps, confirming a hypothesis in the Supporting Info of Yost et al (ref. 5) on the interpretation of earlier tr-2PPE signal by Chan et al. Moreover, the finding suggests the dominance of electronic coupling, at least in the pentacene system, and argues against another proposal of conical intersection (ref. 6 & 7). I congratulate the authors for the success in carrying out this brilliant experiment. This is a major milestone in

experimental research on singlet fission, since the renaissance of this field 12 years ago. This work should serve to guide future mechanistic research on singlet fission. I strongly support the publication of this work in Nature.

(1.0) We thank the referee for the support and appreciation of our work as well as for the insightful comments and the advice on how to improve our manuscript.

Referee #1: The following are optional comments for the authors to consider.

1) The authors cited Berkelbach et al.'s seminal paper on the proposal of CT mediated singlet fission. One thing the authors may want to clarify is that Berkelbach's proposal is mainly on a super exchange type of involvement of CT states, not the actual population of the CT state as is observed experimentally here. There are a number of papers on the role of actually populated CT states in singlet fission, as summarized in ref. 17. Also, the authors cited ref. 32 in the discussion of orbital characters. Ref. 32 also showed the major CT character of S₁.

(1.1) We thank the referee for this comment. Indeed, Sharifzadeh et al.³⁵ (former 32) also report on the CT character. They define CT character as the probability that electron and hole reside on different molecules and find it to be 48% for S₁. We adapted their nomenclature to introduce our envelope function F_{env} . Our findings agree well with their analysis. Berkelbach et al. do discuss the involvement of CT states for different energetic alignments. They contrast *superexchange* and *sequential* mechanisms for pentacene dimers⁴. In the latter case, CT states are visible as intermediate states. This is incompatible with our observations, since the CT states are mixed into S₁ rather than being an intermediate state.

In the *superexchange* case, CT states are only weakly mixed into S₁ and are practically not visible as an intermediate state. In dimers, the CT character of S₁ remains rather small. The case is different in the crystal⁴². Here, CT states make up a major part of S₁ as indeed it should also do for the tetracene and hexacene crystals.

Berkelbach updated his wording in a subsequent article^{R1} to distinguish **small mixing**, i.e. *superexchange*, for dimers and **physical mixing** as is the case for the crystal.

We hence made the following change to the abstract in the revised manuscript:

Here we use time- and angle-resolved photoemission spectroscopy to observe the primary step of singlet fission in crystalline pentacene and show that it occurs in a charge-transfer mediated mechanism with a hybridization of Frenkel and charge-transfer states in the lowest bright singlet exciton.

Referee #1: 2) In the present system of crystalline pentacene, the S1 state is of predominantly CT character. The authors may want to discuss how general this mechanism is when CT is energetically not resonant with FR. Model systems such as tetracene or hexacene come to mind.

(1.2) We thank the referee for pointing this out. We added the following discussion to the manuscript:

Our analysis offers a new perspective on SF in crystalline tetracene⁴⁵ and hexacene⁴⁶. In the former, CT states lie energetically higher and CT mixing into S1 is thus smaller. Nonetheless, the significant Davydov shift in tetracene indicates that in photoemission, the CT-induced transition S1-1 should be visible. Hexacene features CT states that lie lower than in pentacene. We expect them to lie in between FR and TT states, leading to a similar CT character of S1 and a stronger one of 1TT.

Referee #1: 3) While the coherent mechanism may not be dominant here, the authors may want to consider the following. For an ultrafast process happening on the 70 fs time scale, it is difficult to distinguish dephasing from incoherent population decay. For the momentum map of the X state right after photo-excitation, one can also see features of the 1(TT) state. The authors decomposition procedure clearly yields non-zero intensity for the 1(TT) state at time zero. This suggests ~ 20% coherent formation of 1(TT). After all, if there is sufficient

electronic coupling, quantum mechanics teaches us that the states are mixed within the spirit of uncertainty principle, provided dephasing rate is not overwhelming. The authors may want to add some clarification to this point.

(1.3) In the original manuscript, the momentum map TT showed features of both the singlet satellite (X) and the bitriplet. This is because this map was integrated over the dataset shown in Fig. 4c, resulting in inevitable overlap for the given time and energy resolution. Consequently, there was still ~20 % residual signal from the singlet satellite in the TT momentum map. In the revised manuscript, we improved the analysis by taking the momentum map at a later time delay (500 fs), see Fig. R5 (updated Fig. 2b), such that the singlet exciton population has decayed significantly. As can be seen, the common features that the referee pointed out are not visible in this map and the similarity with the S momentum map is enhanced. The features occurring in X are at the same positions as the features of VBM, except for the feature at (0.7, -1.6) A-1 which is specific to the singlet satellite. The updated downstream figures, see Fig. R6 (updated Fig. 4c-e) show the same dynamics at early times as in the original manuscript, i.e. with non-zero population of TT at $t = 0$. This residual population is at the limit of accuracy of our decomposition (~10 %) and we don't interpret it as a coherent formation of 1TT but rather as an imperfection of the decomposition method.

While we do not retrieve the dephasing time in this experiment, we expect electronic dephasing to occur within 10-20 fs based on measurements of other semiconductors at room temperature, i.e. significantly faster dephasing than SF dynamics.

We included the following discussion in the revised manuscript:

Regarding the quality of the decomposition, it is essential to have a well-defined set of basis maps, with one transition being dominant in each map. In our case, the basis is formed on the one hand by the bitriplet after sufficient time delay, such that there is no mixing with residual singlet satellite. The other basis map is the momentum map X, integrated over an energy-time window wherein the singlet satellite is dominant (see Fig. 2B). Due to the time resolution of the experiment, a small amount of residual bitriplet (<10% of total signal) remains mixed into this

momentum map. Some ambiguities in the decomposition hence remain, such as the artefactual signal in the HOMO-projected DOS at 1.8 eV. A feature that is ambiguous is the HOMO-projected signal at 0.7 eV for which it is unclear whether it encodes a HOMO character of the bitriplet or whether it is an artefact. We further validated the decomposition procedure by investigating the dynamics of features specific to HOMO / LUMO (Extended Data Fig.~7).

Reply to Referee #2 (trARPES)

Referee #2: In the article 'Orbital resolved observation of singlet fission', authors Alexander Neef et al have used time-resolved momentum microscopy to map out the orbital character of the transient state in the process of exciton fission in crystalline pentacene. Thereby, they identify the process of fission as being due to a charge-transfer mediated mechanism, an issue that was still debated in literature. Utilizing the powerful knowledge provided by orbital-resolved measurements, Neef et al seek to provide insightful understanding into a process that is important to various aspects of material science, particularly to photovoltaic efficiency since the singlet fission process has the potential to double the number of excited charge carriers.

Overall, I find this type of study innovative and powerful. I think the additional information that orbital resolved techniques bring to the table, give a nice holistic picture of the process. The authors have clearly been able to go beyond the information obtained so far, which largely provided only the energetic positions of the transient states. I find the manuscript well written. However, I have some major concerns (and some minor concerns/suggestions) that need to be addressed before the manuscript is reasonable for publication:

(2.0) We thank the referee for the insightful assessment and the important comments.

1. I am quite confused by the heart of the data - the momentum maps in fig. 2b. I would have expected some sort of symmetry/repetitive structures to this measurement corresponding to the many Brillouin Zones that are measured. I am surprised that the momentum maps look so 'disordered' in comparing neighboring BZs. My primary concern is that there are experimental artefacts associated with the alignment of the momentum microscope. My alternate concern is that photoemission matrix elements obfuscate the critical physics that they authors are claiming to observe.

a. For the first, can the authors provide convincing data that the transmission function through the microscope of the photoemitted electron from sample to detector is not the primary cause of the measured distribution? Typically, aberrations and imperfections in alignment lead to a transmission probability that depends on k and E . Such errors could severely impact the measured distributions in Fig.2b and also impact the conclusions.

(2.1a) The excited-state momentum maps of molecular crystals indeed appear "disordered" compared to equivalent data from simpler inorganic crystals. However, this is not caused by experimental artifacts, as is evident from Fig. R1 (Extended data Fig. 1), wherein we show the ordered nature of the dataset. We cut through the second derivative of the photoemission data with respect to energy at the valence band minimum, middle and maximum. The extremal points (M-points) are clearly visible. In momentum microscopy, there is always a slight distortion of the momentum axes caused by the transmission function of the instrument, however, this is not the cause for the apparent disorder of the momentum maps.

Fig. R 1| Determining the reciprocal lattice. Momentum maps of the second derivative of the valence bands at three different energies, corresponding to its minimum, center and maximum. The clearly visible extremal points, especially at the VB minimum, make it possible to align the reciprocal lattice with the data.

The low symmetry of the momentum maps is caused by the photoemission matrix elements. They are very asymmetric since the long axis of each of the two distinguishable pentacene molecules is rotated with respect to the surface normal. The two molecules are furthermore rotated with respect to each other along the long axis. In combination with the low symmetry of the pentacene crystal and the experimental geometry (angle of the incoming beams), all symmetries in the ARPES data are lost (as they exist in inorganic systems such as TMDs, graphene, etc.).

In addition, the excited states are localized many-body states, which is why the excited-state momentum maps appear stronger “disordered” compared to the valence bands.

We added a brief mention in the revised manuscript:

Momentum maps of the states are shown in Fig.~2b. These appear rather disordered compared to those of inorganic crystals as a natural consequence of the low symmetry of the pentacene crystal and the slanted alignment of the molecules.

Fig. R 2 | Momentum maps and projected dynamics for the crystal rotated 10° about the surface normal. **a**, Momentum maps of the singlet exciton, the triplet exciton, the satellite X and the valence band maximum. For the momentum map of the ground state the signal is shown at $E - E_{\text{VBM}} = 0.00$ eV, for those of the excited states, the signal was integrated over the following energy and time ranges: S: (1.60 to 2.00 eV | -10 to 140 fs), X: (0.95 to 1.30 eV | -20 to 60 fs), TT: (0.50 to 0.80 eV | 580 to 620 fs). **b**, Momentum-integrated dynamics, equilibrium signal subtracted. **c**, Orbital-projected dynamics with the maps TT for LUMO character and X for HOMO character.

Referee #2: b. What kind of distribution does one expect from a theoretical calculation? It is not too difficult to calculate the photoemission intensity map $I(k, E)$ for the S state and for the HOMO and LUMO (at least). One can also include photoemission matrix elements in such a calculation. Are the authors able to theoretically reproduce the distribution they measure? From the experimental perspective as well, one can rotate the sample or the probe polarization to reduce or symmetrize the effects of the photoemission matrix elements. Can the authors experimentally confirm this?

(2.1b) While we agree that a calculation of our data would be desirable, we must emphasize that such a calculation is out of reach for the current system. Calculating photoemission patterns for the ground states of non-interacting molecules in the plane-wave approximation is possible (see Puschnig et al.³²), but so far, no report has been

published on photoemission patterns of the excited states of interacting molecules as in molecular semiconductors. Other methods are still in an early stage for computing trARPES data (TD-DFT^{R2}) or are too expensive to be applied for the excited states of a crystal with a unit cell as large as pentacene (KKR^{R3}). We hope that our work helps to accelerate these theoretical efforts.

Fig. R2 (Extended Data Fig. 6) shows a new dataset wherein the sample was rotated by 10° about the surface normal with respect to the original dataset. In the resulting momentum maps, the rotation with respect to the original momentum maps is apparent. Yet, they cannot be produced by a simple rotation of the dataset. The operation is more complicated since the angle of the XUV pulses relative to the crystal has changed in the 10° rotated geometry.

Although there is no theoretical explanation of why the excited-state patterns look the way they do, we can assign specific patterns to the involved excitonic states. This is the key aspect of our analysis. Any attempt to symmetrize the data to mitigate the effect of the matrix elements would therefore be counterproductive.

Referee #2: 2. Organic samples are notorious for being damaged during the photoemission process. It would be important to confirm experimentally that the sample isn't being damaged during the process of measurement for the given measurement conditions. Damage to the sample during the measurement process could also be an explanation for the 'disordered' momentum maps. To confirm no damage to the sample, one could continuously measure the sample over a period much longer than the longest experimental window reported in the paper. Then one could confirm that the data accumulated in the first fraction of this measurement matches the data obtained in the last fraction of the same measurement.

(2.2) We do observe a slow degradation of the sample during our measurement, which

Fig. R 3 | Sample degradation. **a**, Degradation of the signal of S, TT and VBM over the course of a measurement. For comparison the total count rate, i.e. all measured electrons, is shown. **b**, Momentum maps of VBM and S of the first 20 minutes of measurement compared to **c**, the last 20 minutes, i.e. 1600-1620 min, of the same dataset as in **a**.

is shown in Fig. R3 (Extended Data Fig. 9). It took 27 hours to take the main dataset presented in this manuscript, during which i) the intensity of the orbital-specific momentum maps continuously decreases and ii) the diffuse signal in-between the momentum peaks increases. This degradation does not, though, change our analysis and finding. This becomes apparent, when comparing the momentum maps in the first (0-20 min) and last (1600-1620 min) twenty-minute interval of the main dataset. Clearly, the degradation blurs the momentum map while leaving the overall pattern unchanged. The degradation is presumably caused by a loss of long-range order of the molecular stacking, which reduces the direct photoemission due to an increase of scattering (inelastic and quasi-elastic) in the photoemission process.

We included a discussion of the sample degradation in the revised manuscript:

We briefly discuss some aspects of our experimental technique. Just like many other organic materials, pentacene experiences light-induced damage when irradiated by XUV light. Over the course of one full data set the VB intensity reduces by 25%, indicating a chemical modification. This leads to a slight blur of the momentum map at the end of the measurement. However, the overall shape and features of the momentum maps are conserved (see Extended Data Fig. 9). While it would be in principle desirable to reduce the degradation, we note that there is a trade-off between acquiring enough signal in the excited states requiring

a long measurement time and reduced degradation requiring a short measurement time (see Extended Data Fig. 10).

Referee #2: (Minor Comments)

3. The authors introduce the phenomenological envelope function $F_{env}(R)$ after discussing the initial wavefunction as a linear combination of Wannier functions w . However, I wasn't able to get the quantitative connection between the two quantities. Is there a way to present the envelope function $F_{env}(R)$ quantitatively?

(2.3) Both ground and excited electronic states in molecular crystals are localized in real space due to structural fluctuations in soft lattices as well as strong electron-hole interactions. Such states can be expressed as superpositions of Bloch states or, equivalently, by a combination of Wannier functions. We introduced the envelope function (inspired by Sharifzadeh et al.³⁵ (former 32)) to describe the localization of electrons. The envelope function (in real space) is related to the width of a state's distribution in momentum space as shown by Puschnig et al, ref. 32, for HOMO states or by Dong et al. (DOI: 10.1002/ntls.10010, especially Fig. 4) for excitons in an inorganic semiconductor. Due to the more complex structure of the excited state maps in pentacene, however, we don't attempt a quantitative analysis of the exciton sizes in the present work.

We made a slight change in the revised manuscript:

Here we additionally introduced the phenomenological envelope function³⁵ F_{env} , which is a distribution function centered at the origin. It modulates the magnitudes of Wannier functions on different unit cells and thereby their contribution to the total wavefunction.

Referee #2: 4. The authors describe that at a fluence of 0.2mJ/cm², the singlet provides about 1% of the VB signal. How does this compare to the known absorption in the material at the pump wavelength?

(2.4) At the pump wavelength 685 nm, $\epsilon_1 = 4$ and $\epsilon_2 = 5$ (Dressel et al.^{R4}). This yields a transmission of 52% across the surface at 65° incidence angle for s-polarized light. With the resulting absorption coefficient $\alpha = 2.0 \times 10^5 \text{ cm}^{-1}$ and the pentacene unit cell volume $V = 2.9 \times 10^{-27} \text{ m}^3$, we get an excitation ratio of 2.5%, similar to the observed 1%. We added this discussion to the Extended Data.

Referee #2: 5. In the manuscript, the authors excite at 1.81eV with polarization along the a-axis of the crystal. It would be helpful to give the context of the measurement, e.g. in order to excite the S state resonantly.

(2.5) We thank the referee for this helpful comment. Indeed, we didn't state clearly why we chose to excite at 1.81 eV. We added a sentence in the revised manuscript.

In the experiment, pentacene is excited with a 1.81 eV pump pulse polarized along the crystal a-axis, thus resonantly populating the lowest bright singlet exciton.

Referee #2: 6. Related to the above question, can the author also comment the dynamics that allow a significant occupation of the X state after resonant excitation to the S state within the ~30fs temporal resolution of the experiment?

(2.6) The signal X is not the signature of a state but rather a satellite peak of the singlet exciton S1 originating from a different molecular final state of the photoemission process. As described in Fig. 3, it derives from the CT character of S1 and features, within the experimental uncertainty, the same dynamics (Fig. 4). We improved our description and now indicate this already in the caption of fig. 2.

Referee #2: 7. As a suggestion, I found the 3D figures in 2A hard to understand. I leave it to the authors to see if this can be improved.

(2.7) We acknowledge that the 3D presentations are quite involved and that it takes a reader some time to visually access the rich data. Yet we wish to keep the data representation as it allows us to display, and the reader to appreciate, the full

Fig. R 4| Non-normalized dynamics. The non-normalized dynamics of orbital-projected S, X and TT.

dimensionality of our datasets since this is essential for our analysis. In our comparison of the momentum maps and the assignment to specific states, we reduce the dimensionality of the data.

Referee #2: 8. In Fig. 4e and f, it would be helpful to see the non-normalized datasets. Also, why does the data for TT terminate at 150fs? What happens after 150fs now that the state is populated?

(2.8) We added the non-normalized datasets to the Extended Data, see Fig. R4 (Ext. Data Fig. 8). In the original manuscript, the TT data terminates at 150 fs because it shows the main (highest-statistics) dataset. The decomposition was not performed for the other datasets shown in Fig. 4f. In the revised manuscript, we replaced the orbital-resolved data in for TT with the non-projected dynamics of the lower-energy signal (thus a sum of X and TT) to display the longer dynamics, see Fig. R6f. After 150 fs, the triplet state remains populated with no appreciable decay up to 30 ps. Furthermore, we also show TT momentum maps at longer time delays (Fig. R6g, also in revised manuscript). After the primary step, we observe no change in the momentum maps.

Reply to Referee #3 (singlet fission spectroscopy)

Referee #3: The manuscript by Neef et al presents advanced time-resolved photoemission spectroscopy study of early singlet fission dynamics in pentacene. The study and results are following closely XYZhu study from 2011 (ref 3), but, in addition to the states' energies, authors record the momentum maps to recover additional information about the states'

Fig. R 5| Comparison of the new and old momentum maps TT.

localisation. Experimental data are unique and well-presented. The timescales of fission and the respective energies of the states agree well with previous studies on pentacene. The key novelty claim by the authors is the observation of CT character in the state 'X' preceding the bi-triplet formation. My key concern with this (experimentally very sophisticated and very well-performed) work is that no strong evidence is provided for CT character of state X. A number of qualitative speculations are put forward, however, no quantitative analysis is given to evaluate the likeliness and the extend of CT contribution. My (may be rather naïve) view is that momentum maps for state X can actually be seen as a superposition of momentum maps of S and TT. This would indicate that X is a mixed S-TT 'hot' state, may be the same as one observed in the analysis of coherent vibrionic dynamics. Such interpretation would not require CT state involvement, except via superexchange

(3.0) We thank the referee for the positive general assessment and for raising this concern. In response, we acquired additional data, especially high-statistics data at longer delays compared to the initial data. We now use data at 500 fs delay to obtain a purer TT map with essentially no residual S1-1 contribution. We realized that the original TT map still contained ~20% residual S1-1 character. The new TT is compared with the old one in Fig. R5. With this new map, we achieved a cleaner decomposition (Fig. R6). The difference between X and TT signals is now more obvious. The HOMO-specific peaks in X are neither found in S nor TT such that a combination of the latter can hardly yield X. Hence, X cannot be a sign of the S-TT state and another explanation

must be sought. The CT-mixing that we propose can explain X as a satellite of the singlet exciton and is consistent with a significant part of theoretical analysis in the SF literature.

We added a discussion of the decomposition and the importance of a good basis to the revised manuscript:

Regarding the quality of the decomposition, it is essential to have a well-defined set of basis maps, with one transition being dominant in each map. In our case, the basis is formed on the one hand by the bitriplet after a sufficient time delay, such that there is no mixing with residual singlet satellite. The other basis map is the momentum map X, integrated over an energy-time window wherein the singlet satellite is dominant (see Fig. 2B). Due to the time resolution of the experiment, a small amount of residual bitriplet (<10% of total signal) remains mixed into this momentum map. Some ambiguities in the decomposition hence remain, such as the artefactual signal in the HOMO-projected DOS at 1.8 eV. We further validated the decomposition procedure by investigating the dynamics of features specific to HOMO / LUMO (Extended Data Fig.~7).

We updated all figures with the data from the decomposition with the new TT map.

Referee #3: I also have a number of more specific comments:

1) figure 2a - the choice of presentation of momentum maps is artistic, but i feel use of 3D surfaces complicates interpretation.

(3.1) We addressed this aspect in reply 2.7.

2) time resolution of the experiment (sadly comparable to fission timescale) is not commented on in the main text, only in SI. I believe it is an important limitation of the method which should be discussed.

(3.2) Our experiment resolves the singlet fission process. As there is no unique definition of 'time resolution' in pump-probe experiments, we now state the system response function, i.e. the cross-correlation of the pump and probe pulses. While the system response function (43 fs FWHM) is shorter than the fission process, we point out

Fig. R 6| Orbital-projected dynamics and evolution at longer delays. **a**, Momentum-integrated dynamics of SF in pentacene, showing the differential PE intensity (equilibrium signal subtracted). **b**, Momentum maps at $E - E_{\text{VB}} = 0.8$ eV and illustration of the decomposition procedure. **c**, The dynamics of states with HOMO and **d**, LUMO character. **e**, Orbital-projected population dynamics of the excited states shown with the model fit. The signal was integrated over the shown energy range in **c** and **d** to reduce spurious counts from lower-lying states. **f**, Dynamics of the singlet and the lower energy signal at longer delays, not orbital-projected. **g**, Momentum maps of the lower energy signal (integrated from 0.6 to 1.2 eV) at longer time delays.

that even exponential dynamics faster than the response function could be resolved by deconvolution techniques.

The non-equilibrium state is then probed by a p-polarized ionizing extreme-ultraviolet (XUV) probe pulse (Fig.~1c, d)³⁸ with a system response function <50 fs.

Referee #3: 3) 1% of excited molecules corresponds to a very high pump fluence. Do authors see any power dependence, particularly after 100 fs where bimolecular process can kick in?

(3.3) Within the investigated time range (up to 30 ps) we did not see any pump-power dependence. The measurement presented in the manuscript was performed at the highest possible fluence before pump-induced photoemission distorts the signal. A significantly lower fluence would lead to unfeasibly longer measurement times since the sample also degrades over time, see Fig. R3 (Extended Data Fig. 9) and Extended Data Fig. 10. We stress nonetheless that the SF time scale we observe is entirely consistent with all measurements performed thus far, spanning a range of excitation

energies and fluences. Compare for example the measurements by Wilson et al.^{R5} with significantly higher fluences (~1 mJ / cm²).

In general, there are only two processes which technically could interfere with our measurements:

- i) Repopulation of the 1TT state by triplet-triplet annihilation is expected on much larger timescales, well beyond 1 ps (see Bossanyi et al, DOI: 10.1038/s41557-020-00593-y).
- ii) Biexcitonic singlet-singlet processes: We can exclude the annihilation of two S1 excitons to a higher S_n exciton (S1 + S1  S0 + S_n) by means of the energy resolution of our experiment of 150 meV. The S2 state, i.e. the higher-lying Davydov component of the crystal is about 120 meV higher in energy than the detected S1 state. As we see no signal at higher energies over the whole time of the experiment, we exclude any significant contribution from this process.
- iii) Singlet- triplet and singlet-1TT processes: Quenching processes to the ground state involving one singlet and one triplet/ triplet-triplet species are hard to distinguish in our experiments. However, fluence dependent fluorescence experiments probing the Herzberg-Teller emission of the 1TT state show that nonlinearities with the pump fluence, expected for bimolecular processes, only occur after about 5 ns (Bossanyi et al, DOI: 10.1038/s41557-020-00593-y). As the formation of uncorrelated triplets occurs on much larger timescales, we also exclude any contribution from singlet-triplet processes.

Referee #3: 4) I find the separation of fission dynamics into three steps just based on the exponential timescales speculative. Are there evidence for clear systematic changes and energies or/and momentum maps on the timescales of 300 and 600fs?

(3.4) After the primary step, we observe no changes in the TT momentum maps up to 10 ps, see Fig. R6g (also in revised manuscript). Yet we do not exclude that minor mixing is still present even at later times, such as a possible CT character of 1TT. In the momentum maps at 90 and 200 fs there is residual S1-1 mixed into the TT map. There are several plausible explanations for the observed biexponential kinetics. With the knowledge from our improved decomposition, we deem it most likely that mixing of

CT into the adiabatic 1TT is the cause of the higher energy signal after 200 fs. We modified the discussion in the revised manuscript as follows:

After the initial 100 fs decay, the signal at 1.81 eV degrades at a substantially slower rate. We deem it most likely that the origin of this second timescale is the separation of 1TT to 1T...T. In fact, the 2 ps timescale for bitriplet separation observed by Pensack et al. in a pentacene derivative is similar to ours (600 fs). The origin of the higher energy signal after 200 fs is then the bitriplet and not the singlet. Such a signal can be readily explained with a mixing of diabatic TT and CT states in the adiabatic bitriplet exciton (see Ext. Data Fig. 11). The CT character, which is much smaller than in the singlet exciton, is then lost in the transition to the non-interacting separated bitriplet exciton. Two more observations make our interpretation likely: (1) the higher energy signal shifts down in energy by >50 meV over the course of the primary step (see Ext. Data Fig. 12) and (2) the residual HOMO character that our decomposition procedure finds in the low-energy signal at 140 fs (Fig. 4c).

Referee #3: 5) the energy of state X is exactly half of the singlet energy - is it a pure coincidence? Can this point towards the validity of 'simple' mixed hot S-TT state interpretation?

(3.5) We wish to highlight that the S1 state is a hybrid state with mixed FR and CT character. The signal X is thus a satellite peak of the S1 state and a signature of the singlet's CT character, but not a state by itself. This interpretation is internally consistent with the similarity of X and VBM momentum maps, highlighting the HOMO character of the S1-1 transition causing the signal X. If the referee's speculation of a hot S-TT state was correct, the signal X would have LUMO character which is not the case.

The X signal's energy is not coincidentally near the triplet signal but is expected to be in close proximity, as both signals share the same cationic final state of the photoemission process, see Fig. 3.

Referee #3: Overall, I feel this is experimentally very strong study, however the interpretation of observing CT contribution is not based on solid evidence. May be some alternative explanation can brought forward and discussed. Taking the innovative character of the work but the limited new photophysical insight, it would be well suited for Nature Communications after the above comments are addressed

(3.6) Here, we respectfully disagree with the referee. Our data provides direct evidence of the CT character of the S1 state through its pattern in the momentum maps previously not accessible. We hope that the improved analysis of the revised manuscript convinces the reviewer.

This observation represents a major finding by itself as the CT character of the singlet has been controversial for many years, as is acknowledged by other referees. The CT character underlines a fundamentally different photophysical mechanism of SF in pentacene, thus resolving controversy in the SF community that has persisted for more than a decade. Furthermore, our results show the first images of excitons in molecular crystals and thereby deepen the understanding of these ephemeral states. We are convinced that the proof-of-concept character of our study will spark new investigations in many different fields where intimate knowledge about the wavefunction is crucial. We, therefore, do not doubt that our results will be of interest to a broader scientific community.

Reply to Referee #4 (singlet fission spectroscopy)

Referee #4: Neef et al report trARPES measurements on one of the archetypal singlet fission systems, crystalline pentacene. The mechanism of ultrafast singlet fission in this system has been grounds for extensive debate since 2010, with the most prominent models invoking electronic coherence, vibronic effects/conical intersections, and mediation by coupling to charge-transfer states (typically, virtual CT state in a superexchange pathway). The CT mechanism was posed against models invoking direct coupling between the Frenkel-exciton S1 state and TT, which were

latterly discarded as being unrealistic in the solid. On the experimental side, following the surprising report of coherent fission with an instantaneously populated TT state, later work coalesced around the vibronic models with interpretations grounded in the CT-mediated mechanism proposed by theory. Importantly, the optical spectra used to assign states in these earlier studies and the lack of control over CT states in films meant that it has never been possible to directly experimentally verify the involvement of CT in the solid state (in contrast to intramolecular singlet fission, where the model of CT mediation has been indirectly and directly verified, e.g. Busby, Nat Mater 2015; Margulies, Nat Chem 2016; Alvertis, JACS 2019). Nor have any subsequent models been able to directly explain the signatures assigned to coherent fission in 2011. In this context, the authors' new approach of utilizing momentum- and energy-resolved data to decompose the signals on the relevant ultrafast timescale presents a powerful advantage. According to the authors' analysis, the technique permits direct identification of the initial photoexcited state as a mixture of Frenkel singlet and charge-transfer excitations, as previously determined by steady-state spectroscopy (Yamagata, J Chem Phys 2011). This coherent mixed state evolves directly into TT on the timescale known from optical spectroscopy. To the extent the analysis is robust (see below), this result reconciles the early 'coherent' observations with current theories and definitively validates the CT-mediated mechanism. It is sure to be of great interest to the singlet fission community and highlights the great potential of trARPES to understand other complex ultrafast processes.

(4.0) We thank the referee for this profound assessment and the appreciation of the value of our work.

However, much of the analysis hinges on the claim that the signals can be cleanly decomposed into HOMO-like versus LUMO-like contributions, allowing a decomposition into the X versus S and TT states. From the data provided, it's not clear this is fully justified. In the 1TT momentum map (Fig 2b), the authors highlight the feature in the upper left as characteristic. But stronger features are found in the lower right, including three detectable peaks that appear at precisely the same momenta in the X map. In Fig 4b, I struggle to see any features in the X map that are not present in the TT map, so how can they be cleanly decomposed? What is the meaning of X-like/LUMO-like features in the TT momentum distribution? Is it an oversimplification to project the detected momentum distributions onto these pure diabatic states? What would be the consequence of mixed states (for instance, 1TT is widely held to exhibit both CT and S1 wavefunction character)? As described in the text, this method of separating contributions based on energy and momenta is compelling, but the details cause some concern.

In this regard, the authors claim that "the crucial difference between the [S1 and TT] state is their degree of delocalization", which implies the principal difference in their momentum maps should be the linewidth. Yet there is a strikingly different intensity distribution as well, with many additional peaks in the bottom and right of the TT map, suggesting other differences in the orbital character that are not addressed or explained by the schematic in Figure 3.

Fig. R 7| Dynamics of HOMO and LUMO features. **a**, Selected features characteristic of HOMO and LUMO character on the momentum maps of VBM, X, S, and T (the same as in Fig. 2 of the main text). **b**, Dynamics of signal integrated over all momenta, shown here for comparison. **c and d**, Dynamics of signal integrated over the squares around the characteristic features A and B.

(4.1) We thank the referee for raising these points, which partially overlap with concerns raised by other reviewers. In the revised manuscript, we provide additional data and a refined decomposition analysis. The new analysis is based on taking the TT reference map from new high-statistics data at 500 fs after excitation, in contrast to a pump-probe delay of 150 fs in the original manuscript. In the latter, there was still a substantial amount (~20%) of S1-1 mixed into 1TT. The features that the referee highlights in the TT map are therefore signatures of the satellite of a not yet fully depopulated singlet. With the new momentum map TT, the similarity between S and TT is visually more obvious. The additional peaks that the referee mentioned are not visible, since at 500 fs the singlet is significantly less populated and the S1-1 contribution is below the uncertainty of the decomposition method.

Two notes at this point: i) The decomposition method in the original manuscript, which utilizes the full momentum information, seemed to be more robust against the S1-1 admixture compared to the human eye-inspection of the maps. ii) The relative brightness of the main features in the TT map is slightly different than for TT and S in the original manuscript. This is due to a slightly different surface alignment on the measurement spot on which the momentum map at 500 fs was acquired.

In the revised manuscript, we relate our approach to theoretical work that introduced the idea of decomposing momentum maps of molecular ARPES data⁴⁴. While these authors used computed momentum maps as a basis set, our approach extends the idea by using an intrinsic basis from the experiment with a non-trivial geometry and by

investigating excited states. As the referee has noticed the quality of the basis is important. With the new data and analysis, we hope that the reviewer is convinced that we now do have access to a well-defined basis within a single measurement.

To further convince the referee of the stability of the decomposition, we performed an additional analysis, see Fig. R6 (Extended Data Fig. 7). Here we integrated the signal over a small region around some characteristic features. The resulting momentum-integrated dynamics are then sensitive to HOMO / LUMO character and are akin to the dynamics obtained by orbital projection.

We added the following discussion to the revised manuscript:

Regarding the quality of the decomposition, it is essential to have a well-defined set of basis maps, with one transition being dominant in each map. In our case, the basis is formed on the one hand by the bitriplet after sufficient time delay, such that there is no mixing with residual singlet satellite. The other basis map is the momentum map X , integrated over an energy-time window wherein the singlet satellite is dominant (see Fig. 2B). Due to the time resolution of the experiment, a small amount of residual bitriplet (<10% of total signal) remains mixed into this momentum map. Some ambiguities in the decomposition hence remain, such as the artefactual signal in the HOMO-projected DOS at 1.8 eV. A feature that is ambiguous is the HOMO-projected signal at 0.7 eV for which it is unclear whether it encodes a HOMO character of the bitriplet or whether it is an artefact. We further validated the decomposition procedure by investigating the dynamics of features specific to HOMO / LUMO (Extended Data Fig.~7).

Regarding the reviewer's comment about a possible mixed-state character of the 1TT state, we appreciate that there might be CT character in 1TT. In fact, this could be the deciding difference between 1TT and 1T...T. Yet, we emphasize that such CT mixing will be small because of the substantial energy difference between diabatic TT and CT. This contrasts with the almost isoenergetic CT and FR in pentacene. Evidence for [TT<>CT] could come from transitions that are not possible in pure TT and arise from the CT character (see Ext. Data Fig. 11). These are transitions from the LUMO of 1TT and leaving D0+ behind. This transition would appear at the total excitation energy of

1TT (1.72 eV) in line with a slight reduction of the higher-energy signal that we observe (see Ext. Data Fig. 12). Furthermore, a transition from the HOMO to D1+ would also be possible. This transition would be isoenergetic to the main transition seen in Fig. 3 and have HOMO character. Within this interpretation, we would expect to see HOMO features in the TT map at 500 fs which we do not observe. This does not totally exclude CT character in adiabatic 1TT, since the CT character might be too small to be visible. The above interpretation would also be consistent with the biexponential decay of the higher energy signal (Fig. 4f), implying a contribution from the 1TT>D0+ transition as a signature of the bitriplet state. In fact, the second decay rate seems likely to be the time scale of bitriplet separation. We updated our manuscript with this interpretation:

After the initial 100 fs decay, the signal at 1.81 eV degrades at a substantially slower rate. We deem it most likely that the origin of this second timescale is the separation of 1TT to 1T...T. In fact, the 2 ps timescale for bitriplet separation observed by Pensack et al. in a pentacene derivative is similar to ours (600 fs). The origin of the higher energy signal after 200 fs would then be the bitriplet and not the singlet. Such a signal can be readily explained with a mixing of diabatic TT and CT states in the adiabatic bitriplet exciton (see Ext. Data Fig. 11). The CT character is then lost in the transition to the non-interacting separated bitriplet exciton. Two more observations make our interpretation likely: (1) the higher energy signal shifts down in energy by >50 meV over the course of the primary step (see Ext. Data Fig. 12) and (2) the residual HOMO character that our decomposition procedure finds in the low-energy signal at 140 fs (Fig. 4c).

For completeness, we also add that even if the wavefunctions of S1 and 1TT are of the same orbital character, the corresponding momentum maps would show slight differences for two reasons: (1) the photoemission probes a hemispherical cut through the Fourier-transformed orbital where the radius of the sphere is inversely proportional to the square root of the ionization energy of a state (for fixed photon energy). Since S1 and 1TT have different ionization energies, the photoemission process (PE) selects slightly different cuts of their respective wavefunctions³². (2) We introduced the PE process as a Fourier transform in Fig. 1. Extending on this picture, it becomes clear that the more delocalized a state of the same orbital character is, the more the PE intensity

shifts to the k vector of that state. In other words, electron delocalization localizes the intensity in momentum space to narrow peaks, whose intensity is modulated by the orbital character. Therefore, the width of the peaks of the momentum distribution can also provide additional information about the localization of (excited) molecular states, which e.g. can be seen in the comparison of the valence band and the state X, where the latter one is clearly broader in momentum space, i.e. more localized in real space.

We updated all figures with the data from the decomposition with the new TT map.

Referee #4: In the conclusions, the authors nicely explain how their observations rule out the original coherent model. But it is not clear how the argumentation in lines 175-184 relates to the observations in the present work, or on what basis it rules out the involvement of a vibronic mechanism. The model is agnostic to the nature of the initial singlet state, and it remains entirely compatible with the kind of CT mediation invoked here (indeed, some of the same authors explicitly invoke CT-mediated coupling within the SI scheme in subsequent work, Schnedermann et al., Nat Commun 2019). If the authors are arguing that the vibronic mechanism is incorrect, can the present model account for the presence of coherent vibrations reproducibly observed in the SF product state TT? It seems that the present study demonstrates a powerful tool to understand the nature of the states involved in the singlet fission pathway - an important advance - but not the relevant couplings that drive the process. Momentum resolution yields a more detailed assignment of states, but the end result is still a kinetic model of the form $A \leftrightarrow B \leftrightarrow C$. The rate constants are purely empirical and it still falls to theory to map these onto matrix elements.

(4.2) We do not rule out the involvement of vibrations and regret that our argumentation in the original manuscript caused this impression. Indeed, vibronic coupling is essential to singlet fission as the surplus energy generated by the process in pentacene must be

taken up by vibrations. Otherwise, the system would show Rabi oscillations (electronic coherence) as discussed by Berkelbach^{R6}.

Also, there is unambiguous experimental evidence in a number of papers that singlet fission is accompanied by coherent vibrations in a wide frequency range. These are launched by displacive excitation^{R7} either in the S0-S1 transition or in the S1-TT transition. While the initial optical excitation (S0-S1) may trigger all frequencies, high-energy coherent vibrations with oscillation periods $< \sim 50$ fs are too fast for being triggered by the S1-TT transition, as already discussed elsewhere^{R8}. These high-frequency modes survive the singlet fission process and are spectroscopically visible on the triplet exciton PES. A conical intersection is therefore not necessary to transfer the vibrational coherence to the TT state.

The question we are addressing is whether a conical intersection is a necessary condition for singlet fission, which we argue is not the case. First, we rule out the involvement of slow, intermolecular vibrations in a CI mechanism. We do not observe a time scale neither by oscillatory kinetics or the overall SF kinetics that would fit such slow modes as claimed by Duan et al.^{R9}. Furthermore, CIs are commonly constructed by the combination of a tuning and a coupling mode. Kinetics dominated by a CI therefore necessitate weak coupling in the equilibrium geometry which might be the case in a pentacene dimer but is not the case in the crystal^{R6, R10}. Specific modes might change the coupling and thereby accelerate (or slow down) singlet fission, but this is not the dominant mechanism. Coupling to an incoherent phonon bath is sufficient to explain the SF timescale^{R10}.

Our technique reveals the nature and dynamics of the electronic states. The electronic states themselves encode a major part of the coupling necessary to induce singlet fission. Strong coupling between CT and FR states mixes these states in the lowest bright singlet exciton and the amount of mixing is directly related to the electronic coupling strength in the form of matrix elements between diabatic states. This new “mixing observable” can be used to verify computations as done by Sharifzadeh et al.^{R11} and several others. As argued above, our results also hint at mixing between CT and TT which would then reveal all relevant electronic couplings.

In conclusion, we claim that, while conical intersections might be present and even slightly speed up singlet fission, they are not a necessary condition for singlet fission in crystalline systems.

We replaced the previous discussion on a conical intersection mechanism with the following:

Our observations show the nature of the electronic states and reveal the significant CT character of the singlet exciton. They do not reveal the nuclear rearrangement that follows the perturbation by optical excitation and singlet fission. The nuclei will relax in both steps. Specific vibrational modes will be launched that shuttle the wavepacket to lower energies. As such, our observations do not exclude a vibronic or a conical intersection mechanism based on high-frequency modes and are consistent with the observation of specific vibrational modes in the SF product states. At the same time, our observations do not provide any evidence for conical intersection dynamics, i.e. as a modulation of the fission rate with the vibrational frequency. Mechanisms that do not rely on the assistance of specific vibrations but rather on the coupling to a phonon bath are sufficient to explain singlet fission dynamics in crystalline systems.

Referee #4: In addition to addressing these points, the authors should update their framing of the CT-mediated model of singlet fission to note it has been substantially validated in intramolecular systems.

(4.3) We adopt this suggestion and updated the revised manuscript as:

In systems that exhibit intramolecular SF, the CT-mediated mechanism has been substantially validated in solution²⁷⁻²⁹.

Minor points:

-I believe the ratio of [TT]/[S1] should be 5, rather than 0.2.

(4.4) Corrected. Thank you for spotting this mistake.

-I could find no discussion of statistics or error bars. While the data looks clear, there should be a discussion of signal vs background counts and noise levels.

(4.5) We addressed parts of this comment in 4.2 and added the following discussion to the manuscript:

We briefly discuss some aspects of our experimental technique. Just like many other organic materials, pentacene experiences light-induced damage when irradiated by XUV light. Over the course of one full data set the VB intensity reduces by 25%, indicating a chemical modification. This leads to a slight blur of the momentum map at the end of the measurement. However, the overall shape and features of the momentum maps are conserved (see Extended Data Fig. 9). While it would be in principle desirable to reduce the degradation, we note that there is trade-off between acquiring enough signal in the excited states requiring a long measurement time and reduced degradation requiring a short measurement time (see Extended Data Fig. 10).

We also added a discussion of the uncertainty in the kinetic fit to the Methods section.

Referee #4: -In extended data Fig 3, what is the justification for the choice of peaks used for linewidth analysis? In the TT and X maps, other and stronger peaks appear to be available, with what seem by eye to be different widths. How representative/meaningful is the parameter currently extracted?

(4.8) We chose the peaks for their isolation and their hence clearly definable FWHM. This is more evident in the updated TT map with no S1-1 residual. The parameter can be used for a rough comparison of peak widths. We added MDCs at more locations in the momentum maps to this figure (Extended Data Fig. 4).

Additional remark:

In the original manuscript there was a minor inconsistency in the assignment of the reciprocal and the corresponding real lattice. This is corrected in the revised manuscript. We furthermore made slight aesthetic changes.

References:

[R1] T. C. Berkelbach, *Advances in Chemical Physics* **162**, 1-38 (2017).

[R2] S. Hammon, S. Kuemmel, *Phys. Rev. A* **104**, 012815 (2021).

- [R3] H. Ebert et al., *Rep. Prog. Phys.* **74**, 096501 (2011).
- [R4] M. Dressel et al., *Optics Express* **16**, 19770-8 (2008).
- [R5] M. W. B. Wilson et al., *J. Am. Chem. Soc.* **133**, 11830-3 (2011).
- [R6] Berkelbach et al., *J. Chem. Phys.* **138**, 114102 (2013).
- [R7] Zeiger et al., *Phys. Rev. B* **45**, 768-78 (1992).
- [R8] Musser et al., *Nature Phys.* **11**, 352-7 (2015).
- [R9] Duan et al., *Sci. Adv.* **6**, eabb0052 (2020).
- [R10] Refaely-Abramson et al., *Phys. Rev. Lett.* **119**, 267401 (2017).
- [R11] Sharifzadeh et al., *J. Phys. Chem. Lett.* **4**, 2197-201 (2013).

Reviewer Reports on the First Revision:

Referee #2 : This referee did not write a "Remarks to the Author", but mentioned to me (Yohan) having been through the authors' responses, and the referee is satisfied and supportive of publication.

Referee #3 (Remarks to the Author):

The revised version is a clear improvement over original manuscript. Extra datasets, more quantitative analysis, and particularly careful response to reviewer 2 and 4 comments made the findings much more robust.

I would still ask the authors to look at the following issues:

- the discussion in the main text is now more balanced, however in a few places including the abstract it is stated that "singlet fission in crystalline pentacene ... occurs in a charge-transfer mediated mechanism" . I would be more careful and emphasise that experiment provides evidence for CT character of S state (or rather optically accessible bright mixed state). There are no experimental evidence provided on the role this CT character plays in fission.

- Experimental resolution is claimed to be ~ 45 fs. However, it is probably lower taking the experimental risetime in figure 4f is clearly longer than in the model curve. With this in mind it may not be fully appropriate to make make claims about the initial (direct, coherent) TT population. In contrast to what authors state, it may be non-zero and experiments with higher time resolution are required to quantify the contribution of TT to optically generated states.

Referee #4 (Remarks to the Author):

The authors have thoroughly addressed my concerns and, in my view, those of the other reviewers. The new decomposition looks much cleaner, and the conclusion regarding the CT contribution is much clearer (and more clearly framed). This is an exciting advance, and I enthusiastically support publication.

Author Rebuttals to First Revision:

Reply to referees' comments

Throughout this reply comments by the referees are marked in grey Courier font and answers by the authors in blue. Quotes from the revised manuscript are indented, *italic* and in black.

Reply to Referee #3

Referee #3: The revised version is a clear improvement over original manuscript. Extra datasets, more quantitative analysis, and particularly careful response to reviewer 2 and 4 comments made the findings much more robust.

(3.0) We thank the referee for the appreciation of our efforts in the revision. We agree that the comments by the referees helped to enhance the manuscript and make it more convincing.

Referee #3: The discussion in the main text is now more balanced, however in a few places including the abstract it is stated that "singlet fission in crystalline pentacene ... occurs in a charge-transfer mediated mechanism". I would be more careful and emphasise that experiment provides evidence for CT character of S state (or rather optically accessible bright mixed state). There are no experimental evidence provided on the role this CT character plays in fission.

(3.1) The line of argumentation is that the observation of CT states suggests a CT-mediated mechanism in pentacene. All presented observations are consistent with this mechanism, which is not the case for the other mechanisms. However, we appreciate the viewpoint of the referee, in particular as we report the first direct observation of the involvement of CT states in crystalline pentacene. We changed the wording at the respective places in the manuscript to reflect a more nuanced tone:

~~...and show that it occurs in a charge-transfer mediated mechanism with a hybridization of Frenkel and charge-transfer states in the lowest bright singlet exciton. (old version)~~

Our results suggest a charge-transfer mediated mechanism with a hybridization of Frenkel and charge-transfer states in the lowest bright singlet exciton. (new version)

In the discussion:

~~The direct evidence of physical mixing of CT-states into S1 thus validates the purely electronic CT-mediated mechanism. (old version)~~

The direct evidence of physical mixing of CT-states into S1 is thus perfectly consistent with the purely electronic CT-mediated mechanism. (new version)

Referee #3: Experimental resolution is claimed to be ~45fs. However, it is probably lower taking the experimental risetime in figure 4f is clearly longer than in the model curve. With this in mind it may not be fully appropriate to make make claims about the initial (direct, coherent) TT population. In contrast to what authors state, it may be non-zero and experiments with higher time resolution are required to quantify the contribution of TT to optically generated states.

(3.2) The instrument response function (IRF, i.e. the pump-probe cross-correlation) is not the same for all data sets. The 'main data set' was taken with an IRF of ~45 fs (obtained by fitting the model), while the data sets for longer time delays were taken with a slightly different experimental configuration resulting in an increased IRF. We thank the referee for this comment and now clearly describe this fact in the supplementary materials and the caption / legend of Fig. 4.

We stress that the same model is shown in Fig. 4e and 4f, obtained by fitting the main data set. This fit is consistent with the data. Since the data sets for longer time delays are only cited to show the bi-exponential decay of the signal, the different time resolutions of these data sets do not affect the conclusions concerning an instantaneous TT population.

In addition, we note that the term ‘time resolution’ is usually ill-defined. An IRF of 45 fs FWHM does not imply that the experiment is blind to any dynamics faster than this value. Given good statistics and appropriate analysis, also process characterized by a time constant shorter than the IRF can be revealed. On this basis, we can exclude a *significant* quasi-instantaneous TT population while the statistical uncertainty of our experimental data and analysis (specifically in the decomposition) cannot fully rule out a minor instantaneous TT contribution. A key conclusion of our work is therefore that the coherent mechanism is not governing the singlet fission process.

We slightly modified Fig. 4f to clearly distinguish the data sets and added the following sentence to the legend of Fig. 4f

The short data set and model fit are the same as in e. The long data sets were acquired with a slightly longer instrument response function.